# Structural Changes in the Romanian Economy Reflected through Corine Land Cover Datasets

**Alexandru Rusu, Adrian Ursu \*, Cristian Constantin Stoleriu, Octavian Groza, Lilian Niacșu, Lucian Sfîcă, Ionuț Minea and Oana Mihaela Stoleriu** 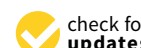

Faculty of Geography and Geology, Alexandru Ioan Cuza University, Iași 700506, Romania; alexrusucuguat@yahoo.com (A.R.); cristian.stoleriu@uaic.ro (C.C.S.); octavian.groza@uaic.ro (O.G.); lilian.niacsu@uaic.ro (L.N.); sfical@yahoo.com (L.S.); ionut.minea@mail.uaic.ro (I.M.); oana.stoleriu@uaic.ro (O.M.S.)

\* Correspondence: adrian.ursu@uaic.ro; Tel.: +40-723-246-639

**Abstract:** During the last 30 years, the Romanian economy has faced different challenges due to structural readjustments, overcoming crisis and globalization. The share of primary and secondary sectors in the gross domestic product have strongly decreased, while the services have taken off. The main objective for this study is to observe how these economic readjustments can be assessed and measured using the Corine Land Cover datasets from 1990, 2000, 2006, 2012 and 2018 (with special observation on the range 2006 and 2018, after Romania was included in European Union). Despite some of the methodological limitations (like the minimum surface change), the Corine Land Cover turned out to be a powerful tool and it allowed us to detect an intense correlation between the socioeconomic and the structural trends in land use, in specific spatial contexts. The artificial surfaces are constantly increasing and this trend is rather visible as a distance function to the major Romanian cities. The most interesting changes occurred in the case of the agricultural polygons. The main trend emphasized by our analysis regards the redeployment of large farms in areas of agronomic and environmental territorial optimum. Such is the case for vineyards (after a decline during 2000–2006) and for annual cultures. All these changes in land-use patterns are too complex to be encompassed by a single methodology, which is why we used different tools, ranging from spatial analysis to geo-economic modeling, in order to detect how the Corine Land Cover datasets might be used for a better understanding of the Romanian economic readjustments.

**Keywords:** Corine Land Cover; structural changes; land use; Romania; data inconsistency

## 1. Introduction

After 1990, during the post-communist period, Romania passed through complex social, economic and political transformations which led to radical background changes within some important fields such as land property and agricultural land exploitation. Some of these transformations were investigated at global [1], European [2] and national levels using Corine Land Cover (CLC) datasets [3–5]. The CLC project largely fulfills the expectations of researchers and other users [6], having two main advantages: it provides a seamless geometry usable for macro-regional studies and an internal classification of the land use/cover categories that allows time traceability of changes, for a reasonable period (1990–2018). The facility of tracking the changes in the use of the land has allowed a multitude of applications and correlations between CLC datasets and different parameters [7–9]. There is an intense use of the CLC datasets for local or regional studies; however, the intermediate scales are systematically neglected by other researches [10–12]. This is why we consider that the proper scale of CLC use for policy design should be an intermediate one, an administrative geometry that fits the

needs of analysis somewhere between the local level and the Nomenclature of Units of Territorial Statistics (NUTS3) delineation [13].

The use of CLC at a local administrative unit (LAU2) level is largely unadvised [14], as the degree of land-use generalization might interfere with the local patterns. In this case, numerous policymakers will eventually use CLC just as an informative base of interpretation and not as an analytical frame. At the NUTS3 scale, the CLC dataset overlaps national or regional statistical information that, in some conditions, creates data redundancy or impossible comparisons (mainly in the case of the artificial surfaces) [15].

In this paper we try to investigate the land cover and land changes in Romania, using an appropriate scale of analysis, i.e., an intermediate one between the main administrative frame and the local scale. The period we emphasize in our research mainly covers the 2006–2018 time intervals. The methodology we develop is applied at a national scale, but it can be replicated for other states too. The problem of the intermediate scale of analysis is mainly related to its construction. If the studies at a local scale generally involve a limited set of beneficiaries (policymakers), an intermediate scale would mean to aggregate administrative polygons belonging to a larger set of decision takers. One of the intentions of our study was to develop a method of administrative polygon aggregation that can be intersected with the CLC datasets. This intermediate scale of analysis was labeled pseudo-LAU1, where LAU1 stands for local administrative units of first rank. Using spatial analysis techniques and potential accessibility functions [16], we created an operational pseudo-LAU1 geometry that was used in order to collect the land-use and land-cover information provided in the CLC vector database (2006–2018). The building blocks of this operational geometry are the Romanian local administrative units known as LAU2.

The detection of the anomalies in the land-use/cover changes that we propose in this research has its background in a methodological import from the theory of the economic convergence, more specifically the elaboration of a dynamic typology using class transition matrixes [17]. The integration of the CLC indicators in the alternative geometry allows the evaluation of the relative share of each land-use category in a double chronological context (time $t_0$ = 2006 and time $t_1$ = 2018). If a homogeneous classification in time is applied on the data, the possible state of a pseudo-LAU1 spatial unit takes three possible values: stagnation, positive evolution or decline. Strong evolutions between classes can be assimilated to spatial anomalies in the dynamics of different CLC indicators and they can provide a basis for the selection of interesting case studies [18]. The validation and explanation of the eventual anomalies detected by the class transition matrix is realized using specific geo-statistical techniques and remote sensing analysis. The case studies are declined on artificial surfaces, on a selected set of agricultural land-use dynamics (arable land, vineyards), on forests and on some of the natural land cover layers (wetlands and water bodies). For a better contextual understanding of the main trends registered in the land-use dynamics in Romania, supplementary statistical data were collected (new artificial surfaces in rural areas, mainly agro-industrial facilities) and a chronological dataset of the CLC changes was created as a data quality check tool. From a policy perspective, a sound assessment of land-use changes and anomalies in different territorial contexts is a topic of major interest, as the land-use dynamics might interfere with targeted policies applied for rural development, urban sprawl, transportation and environment.

Considering the information described above, the principal objectives of this article are to investigate potential ways in which the CLC dataset can be used for a more in-depth analysis of land-use and land cover transformations, at an intermediate scale, appropriate for policy design and decision processes in Romania.

## 2. Materials and Methods

*2.1. The Main Economic Changes and Land-Use Transformation in Romania after the Collapse of Communism*

The adoption of the new Constitution of Romania on November 21, 1991 reintroduced, by law article 44, the right to private property, and generated the forces that led to the drastic transformations of the spatial organizations inherited from communism. The avalanche of changes opened by law no. 15 on 7 August 1990, regarding the reorganization of state economic units as autonomous bodies and commercial companies. The entire national wealth was transformed into shares and some remaining state property, while the rest was distributed in 1991 as property certificates to all senior citizens on December 31, 1990.

Corroborated with the effects of the laws of the land property [19], the constructions, the cadastre, the urbanism and the housing [20] and the legal circulation and the legal regime of land owners [21], accelerated the spatial dynamics. The phenomena were accentuated with the 2003 amendment of the Constitution, which allowed the acquisition of land by foreign citizens (laws 312/2005, 17/2014). According to the National Bank of Romania [22], the country has quickly become a target for foreign direct investments in 2007. With 366 greenfield projects, Romania ranked first in Central Europe and 8[th] in the world, in the mentioned year. Of the total 51.15 billion Euro Foreign Direct Invesments (FDI) in-flows since 1990, 63% targeted greenfield investments and 37% brownfield investments. Almost 30% of the total FDI stock was invested in the manufacturing industry, 19% in the construction and real estate sector, 17% in trade and 9.3% in financial intermediation and insurances. The most attractive regions are Bucharest-Ilfov (61.2% of the total), Center (11.3%), West (10.1%) and North-West (6.1%). Urban and peri-urban areas were the most affected, quantitatively and qualitatively. The old industrial platforms were replaced or transformed into industrial parks (a total number of 90, with a total area of 3295 hectares) or commercial areas. Concerning the dynamics of the urban and suburban spaces, the major Romanian and foreign retailers currently own over 4040 super/hypermarkets, warehouses and showrooms, usually grouped in large peripheral areas where they are frequently accompanied by other economic activities.

In the same time, the rural areas have been the object of complex dynamics, based on the consequences of the law regarding the land reserves and later subjected to laws concerning the reconstruction of formerly nationalized properties, including agriculture and forestry [23]. The destructive component of these dynamics is spectacularly described by the demolition or abandonment of facilities and infrastructures (greenhouses and solariums, reception bases, silos, irrigation systems) and the radical modification of the communist-planed plantations (deforestation of forests or annihilation of permanent cooperative cultures). The constructive component is subordinated to the strengthening of private property (a result of new property titles) and the increasingly obvious orientation towards the commercial dimension of agriculture (introducing crops for bio-fuel, multiplying agro-food units, warehouses and depots, agricultural car parks). With some significant regional differences, the rural economy in general (industrial parks and platforms, alternative energy production, urban sprawl) was attracted towards new directions. The densification of the rural habitat is also driven by agro-tourism, with over 2800 agro-tourist pensions in 2019 compared to only 400 in 2000, together with 1197 other infrastructures specific to the rural environment, compared to 574 in 1990.

Due to the administrative fragmentation at the local administrative level (2948 communes in 1990 and 3181 at present), all the spatial dynamics induced by these transformations, presented here in an extremely synthetic form, are hardly subject to an existing form of territorial administration and organization. Paraphrasing Bell [24], who said that "the nation-state is becoming too small for the big problems of life, and too big for the small problems of life", we can say that the same is true today for the Romanian administrative system, caught between the demands of European cohesion policy and the myriad of local interests. All unofficial forms of territorial administration tried so far, most under European pressure, such as development regions and metropolitan areas [25], e.g., IDA )Intercommunity Development Associations), growth poles [26], Local Action Groups

(LAG, imposed by the EU rural development program LEADER), European Groupings of Territorial Cooperation (EGTC, EU Regulation 1082/2006), are rather theoretical constructions, subordinated to policy guidelines, with little or no anchorage at all in the Romanian territorial realities [27]. This is the reason why a functional spatial delineation of an intermediate geometry was needed, in order to better capture the largest amount as possible of land-use change anomalies, also providing a potential framework for case studies or more in-depth analysis.

## 2.2. Datasets and Methods

The data we used in this paper were collected from heterogeneous sources. If the CLC vector layers were downloaded from the Pan-European datasets of Copernicus Land Monitoring Services (v.20), the administrative geometry of Romania (the LAU2 delineation) was collected from the GIS Service of EUROSTAT (GISCO). The degree of generalization that GISCO applied on the product is quite high (1:1.000.000), but this level of generalization is not necessarily interfering with the objectives exposed in the introduction of this paper [28]. After clipping procedures, all the layers were projected in the national projection system of Romania (Stereo 70, a stereographic system with a local Datum named Dealul Piscului 1970). Once all the layers were projected, the polygon's surface was recalculated and three basic geometries were obtained: the CLC Dataset for 2006 (Romania), the CLC Dataset for 2018 (Romania) and the Romanian LAU2 geometry. Despite the fact that the use of the CLC data integrated in a LAU2 frame is quite spread, many precautions are necessary at this level of analysis, if one will take into account aspects like the Minimum Mapping Unit (MMU = 25 ha/100 m), the possible Modifiable Areal Unit Problem (MAUP) effect in the local data integration of land use and local trends of economic specialization [29]. In some cases, when combined, these effects create interpretation artifacts: in 2018, 14 LAU2 from 3186 in Romania are completely missing the category CLC112 (discontinuous urban fabric). In the same year, only 13 LAU2 contain the category CLC111 (continuous urban fabric). In the proximity of many Romanian large cities (Bucharest, Iasi and Cluj-Napoca), the spatial extension of the CLC112 class is still labeled CLC 133 (construction sites), even if the suburbanization process ended before 2016. For some rural LAU2 in the East of Romania, polygons identified as water bodies (lakes, CLC512) are non-irrigated arable land (CLC211), in reality. Some new economic activities that have a distinct land-use pattern (usually, renewable energy facilities) are also confused with CLC211. That is the reason why working with CLC datasets at local levels (LAU2) might be feasible for small areas and with minor corrections, but it could be quite challenging when the needs of policy design demand a regional, national or trans-national scale of analysis and planning [30].

For this kind of scale, an alternative approach is needed. First of all, the large number of LAU2 in Romania (3186, now stable) makes the local data integration of the CLC datasets a feasible but complicated problem. Technically, a consistent number of LAU2 have an "archipelago" shape, being multi-part polygons. In some contexts, this topology attribute is an issue, when dealing with land-use dynamics. Moreover, the splitting trends recorded in the LAU2 nomenclature make the reconstitution of time series extremely challenging, especially for the period we analyze (2006–2018) and especially when one will need to compare the CLC local accuracy with the official statistical datasets [31]. The NUTS3 frame is different: it is extremely stable and its average surface and shape exclude some of the CLC data integration problems, mainly the MAUP issue. However, the number of spatial units is reduced to only 42 elements, which makes this frame too general for significant analysis.

The alternative geometry we propose is based on the concept of LAU1, an administrative geometry that Romania is officially missing, but which at policy level the topic is on the short list of administrative reform discussions and speculations. The algorithm we used in order to delineate this functional and operational alternative geometry is derived from location-allocation models (p-median problems) [32]. The constraints of the model are limited:

(1)    The building blocks of an alternative geometry that fills the gaps between the local and the regional level of analysis and data integration are the Romanian LAU2, defined and generalized by the EUROSTAT geometry, via GISCO.

(2)    The pseudo-LAU1 frame that needs to be created will be spatially limited by the NUTS3 boundaries. This means that two neighbors LAU2 placed in different NUTS3 will not be aggregated in a pseudo-LAU1.

(3)    Generate a contiguity lattice between all the LAU2 centroids that respects the NUTS3 territorial belonging constraint. Populate the links between any pair of neighbor LAU2 with an impedance indicator. This impedance is the geo-statistical filter that will allow the aggregation of LAU2 in pseudo-LAU1 spatial units. In order to obtain heterogeneous new spatial units, we have made an option for an impedance indicator based on a potential accessibility function to population [33]. The equation of the accessibility function weights the demographic masses of all the Romanian LAU2 with the inverse Euclidean distance that separates them.

$$P_i = \frac{1}{M_i * M_j * (1/D_{ij})}$$

(1)

where: $P_i$ = potential accessibility function for each Romanian LAU2 *i*; $M_i$ = population of each Romanian LAU2 in 2015, according to the NSI estimation; $M_j$ = population of each Romanian LAU2 *j* that potentially interacts with *i*, in the frame of each NUTS3; $D_{ij}$ = the Euclidean distance that separates *i* and *j*. Once the potential accessibility function was created and transformed in an impedance function, the contiguity lattice was used as a network that connects all the Romanian LAU2. The network dataset can now be used for implementing p-median analysis (location-allocation models), in order to delineate the pseudo-LAU1 geometry. All the Romanian LAU2 were retained in the analysis, both as candidates for an eventual spatial aggregation and demand points in the model. The potential accessibility function acts like a similarity measure between the features [34]. When the values of the indicator are high, the pairs of LAU2 *i* and *j* are aggregated in a new polygon labeled pseudo-LAU1. If the values are low, an alternative solution is found and the LAU2 are directed to another candidate center. Our first option for a potential accessibility function was the result of a basic intention, to avoid the creation of pseudo-LAU1 that is potentially homogeneous from the point of view of the land use. This solution maximizes the chances that the pseudo-LAU1 frame will intersect different CLC layers (artificial, agriculture, natural land cover or water bodies).

(4)    The total number of pseudo-LAU1 to be created is a function of an objective criterion. Different tests were applied and the optimal solution was fixed at 290 spatial units. Tests with more than 290 pseudo-LAU1 objects often create unique polygons that aggregate only one or two LAU2. When the value is lower than 290, the model emphasizes the role played by the large cities in the demographic system, making unequal size objects. This stable layer of alternative geometry is the frame that will be used in order to detect anomalies in the evolution of CLC categories (Figure 1).

(5)    The stable version of the pseudo-LAU1 geometry was intersected with the CLC 2006 and CLC 2018 seamless vector layer datasets. For each CLC category, a summarizing operation was implemented, so that the pseudo-LAU1 objects can be used to assess changes in time. The land-use/cover surfaces integrated in the alternative geometry were recalculated and expressed in $km^2$. This new database offered us the opportunity to check for relevant changes in the land-use patterns, changes that are dependent on the mathematical formalization of the dynamics [35]. Even if some of the dynamics might look spectacular in relative terms (% growth for a CLC category), they are irrelevant as an absolute difference. Moreover, these changes become even more questionable when reported to the area of the pseudo-LAU1 where we focused our analysis. The absolute change is limited at the scale of the pseudo-LAU1 (0.35 $km^2$ in the first case, 0.17 $km^2$ in the

second case). In relative terms and having vineyards (CLC 221-2006) as a basis, the dynamics look consistent (8% and 60%).

(6)   The recent progress in the use of the transition probability matrix for assessing economic convergence phenomena [36] at different scales suggests that, when dealing with time-dependent anomalies of evolution, using this transition matrix approach is a sound way to investigate trends. As the land-use dynamics are not convergent on a steady state, implementing more sophisticated analysis, such as half-life indicators of the CLC categories, is not recommended because the possible Markov chains will not necessarily act in an ergodic state [37]. This is the reason why, in order to encompass as much as possible of the land-use dynamics, the CLC categories aggregated at pseudo-LAU1 scale were classified, for $t_0 = 2006$ and $t_1 = 2018$. The classification approach was different for each CLC category, as the data distributions are different and the transition matrix functions only with classes that are similar in time.

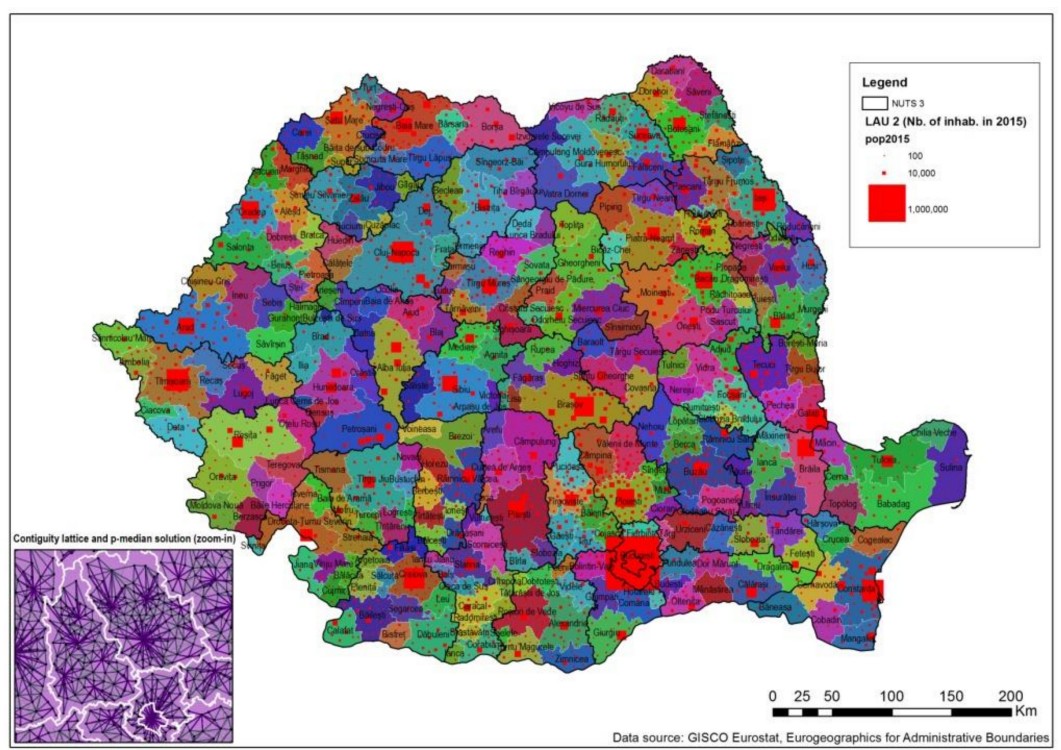

**Figure 1.** The stable version of the alternative pseudo LAU1 (local administrative units of first rank) geometry (290 objects).

Finally, an exploratory statistical analysis was implemented, in order to detect potential statistical associations between the CLC categories in time. The method we retained is the correlation matrix (Pearson's r) between a selected set of variables (14). Strong values of r will indicate that the intensity of land changes is reduced during the analyzed time period. Low values of the Pearson correlation coefficient will mark the CLC categories for which the mutations are strong. The variables were tested for a standard normal distribution pattern (28 Kolmogorov-Smirnov tests) and the risk of rejecting the null hypothesis was lower than 1% in each case, meaning that they are normally distributed.

## 3. Results

### 3.1. From Data Management to Map Output Analysis–Major Trends Affecting Land Use in Romania

The output of the methodological framework that allowed us to create the Pseudo-LAU1 alternative geometry is presented in Figure 1. Following each step described in the methodology, we managed

to create 290 polygons used to intersect the CLC categories in 2006 and 2018. The shapefiles resulted from the intersection were transformed in pivot-tables organized as follows: the rows contain the 290 Pseudo-LAU1 objects, while the columns are reserved for the CLC data.

Once the alternative geometry was created, we started to classify the CLC categories using transition matrixes. The variable we classified is the share of the CLC categories in the total surface of the pseudo-LAU1 geographical objects. The output of a classification is a matrix with rows reserved for the situation of 2006 and the columns for 2018 (Table 1).

**Table 1.** Transition matrix for CLC221 (vineyards) between 2006–2018.

| | | | Share of CLC221 in the Total Surface of Each Pseudo-LAU1 ($t_1$ = 2018) | | | | | |
| | | | 0–3% | 3–6% | 6–9% | 9–12% | 12–15% | 15–18% |
| | | **Class** | **1** | **2** | **3** | **4** | **5** | **6** |
|---|---|---|---|---|---|---|---|---|
| | 0–3% | 1 | **238** | 3 | | | | |
| Share of CLC221 in the | 3–6% | 2 | 21 | **13** | | | | |
| total surface of each | 6–9% | 3 | 2 | 4 | **4** | | | |
| pseudo-LAU1 | 9–12% | 4 | | | | **2** | | |
| ($t_0$ = 2006) | 12–15% | 5 | | 1 | | | | |
| | 15–18% | 6 | | | | 1 | | 1 |

Some other land-use categories might demand different approaches, in terms of relative share classification, but the basic principle remains the same: the detection of anomalies is possible when one pseudo-LAU1 object is shifting 2 classes (positive or negative) between the analyzed period. In the case of Romania, due to a limited manifestation of changes for some CLC categories (both in time and space), a data aggregation of land-use subtypes was necessary, in some cases. Each aggregation was labeled and the quantitative relative indicators resulted were marked in the database (Table 2).

**Table 2.** Correlation matrix between the Corine Land Cover (CLC) categories analyzed (2006–2018). The variables represent the share of each CLC category in the pseudo-LAU1 surface (%).

| | **Variable Labels** | | | | |
| **CLC Categories** | **CLC Data for 2006** | **CLC Data for 2018** | **Pearson *r*** | ***p*-Values** | **Pearson *r* Stability Rank** |
|---|---|---|---|---|---|
| CLC31 | FORr06 | FORr18 | **0.998** | <0.0010 | 1 |
| CLC21 | ARBLr06 | ARBLr18 | **0.997** | <0.0005 | 2 |
| CLC4 | WETr06 | WETr18 | **0.994** | <0.0013 | 3 |
| CLC11 | URBr06 | URBr18 | **0.989** | <0.0001 | 4 |
| CLC12 | INDr06 | INDr18 | **0.982** | <0.0002 | 5 |
| CLC32 | SCRUBr06 | SCRUBr18 | **0.947** | <0.0011 | 6 |
| CLC33 | OPENr06 | OPENr18 | **0.934** | <0.0012 | 7 |
| CLC131 | MINEr06 | MINEr18 | **0.930** | <0.0003 | 8 |
| CLC231 | PASTr06 | PASTr18 | **0.927** | <0.0008 | 9 |
| CLC221 | VINEr06 | VINEr18 | **0.872** | <0.0006 | 10 |
| CLC222 | ORCHr06 | ORCHr18 | **0.867** | <0.0007 | 11 |
| CLC24 | AGRHETr06 | AGRHETr18 | **0.847** | <0.0009 | 12 |
| CLC5 | WATr06 | WATr18 | **0.782** | <0.0014 | 13 |
| CLC133 | CONr06 | CONr18 | **0.762** | <0.0004 | 14 |

In the case of Romania, before implementing the methodological approach based on transition matrixes for the CLC dynamics between 2006 and 2018, an investigation of the CLC changes, from a statistical and technical point of view, was needed. From the statistical perspective, taking into account the MMU size and the aimed accuracy (>85%), with data integrated into the pseudo-LAU1 alternative geometry, the eventual changes can be better understood when a basic correlation matrix is applied [38]. According to the results at pseudo-LAU1 scale, the most stable CLC categories are CLC31 (forests), followed by arable land (CLC21) and wet areas (CLC4). Lower values of the Pearson r are linked with basic categories, like construction sites (CLC133), permanent cultures (CLC221 and CLC222) or aggregated data with specific land-use patterns (water cover, CLC5, and heterogeneous agricultural areas, CLC24). Other CLC categories present intermediate values, but also with very high Pearson r values.

In the case of arable land (CLC221), for example, the intensity of the correlation is so high that it lets few opportunities for founding anomalies in the evolution of this category, based only on a statistical approach (Figure 2). However, when a transition matrix is implemented, at least four possible anomalies (2 classes shift) are detected. This is also the case for other CLC categories and consequently, applying this methodological framework, it will allow us to provide significant investigations on the changes, under the form of specific targeted case studies. Arable land, vineyards and wet area CLC categories were chosen for detailed analysis due to their high sensitivity to socio-economic changes.

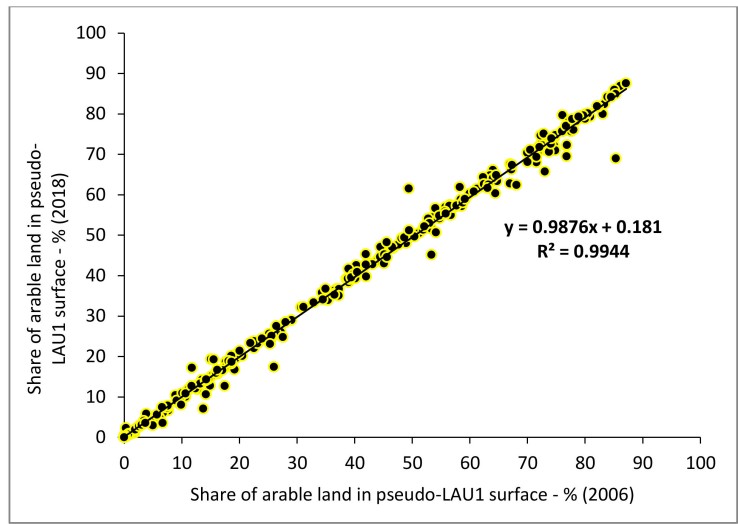

**Figure 2.** Correlation between arable lands relative surface in each pseudo-LAU1 (2006–2018)

Delineating the anomalies in all the CLC changes between 2006 and 2018 is out of the scope of this research article and we have made an option for a data aggregation by some major land-use categories, in order to better investigate eventual mutations. The level 2 urban fabric (CLC11) and the level 2 industrial, commercial and transport units (CLC12) were maintained compact, but CLC131 (mine sites) and CLC133 (construction sites) were included at level 3. For the agricultural areas, the permanent crops were disaggregated for a level 3 analysis (CLC221 and CLC222, vineyards and fruit trees or berry plantations), while all the other categories were included on a level 2 basis (arable land, heterogeneous agricultural areas and pastures—CLC21, CLC23 and CLC24). The same aggregation was applied for CLC31, CLC32, CLC33, CLC4 and CLC5. The reasons for maintaining basic level 3 or aggregated level 2/level 1 category are not only technical, meaning a more rapid implementation of transition matrixes, but are also a matter of territorial scarcity of some land-use types, like CLC112 or CLC213 (rice fields). Our initial hypothesis and research strategy tried to connect economic and social changes with potential mutations in the land-use patterns, at a local scale. However, some of the first investigations on the CLC changes rapidly proved that the main drivers of economic evolutions in Romania might have an impact with intensities lower than previewed. In the same time, having in mind that the utility of the CLC datasets for policymakers should overlay an indicative frame of potential decisions, we considered that a classification of the potential anomalies in the field of land-use dynamics could associate economic drivers and policymakers' expectations. The use of transition matrixes for change assessment is not a new practice in the research field of land use. Our contribution, from this point of view, mainly consists of introducing a new angle of approach, considering that a qualitative delineation of these anomalies is much more supportive for policy decisions.

The strategy followed by our research aimed at detecting eventual anomalies, considering that the irregular changes would be a topic of interest for policy design, having a potential a posteriori economic, social or political explanation. This detection needed the elaboration of a set of transition matrixes for each CLC category describing the agricultural areas and the construction of a qualitative classification of the results. For some of the CLC2 analyzed category, case studies investigating the changes were

implemented. In the second step of our research approach, this method was also implemented for CLC11, CLC4 and CLC5, without providing any qualitative typology because these classes were distinctly aggregated at level 1.

We mainly focused our methodological approach on the transformations registered in the agricultural areas, a stake for policy strategies, if one will take into account that Romania is ranked 5th as an agricultural actor in the European Union. After the elaboration of transition matrixes for each CLC2 category (permanent crops disaggregated by level 3 and arable land, heterogeneous agricultural areas and pastures aggregated by level 2), a qualitative typology of the anomalies was created. If a positive anomaly was detected for an agricultural land-use category, a + symbol was assigned to the pseudo-LAU1. In the case of stagnation in the land-use evolution, the chosen symbol was S. For situations of decline, a – was used to flag the involutions in the database. We define positive or negative anomalies as a shift of at least two classes in the transition matrixes, between 2006 and 2018. The construction of the qualitative typology emphasizes the fact that, at local pseudo-LAU1 level, the agricultural land-use dynamics are more likely to stay stable in time, the situations of decline or evolution being visible only in 35 cases from 290 (12%). No pseudo-LAU1 faced a complete agricultural pattern shifting, the most dynamic ones presenting stagnation for at least two CLC categories (Figure 3).

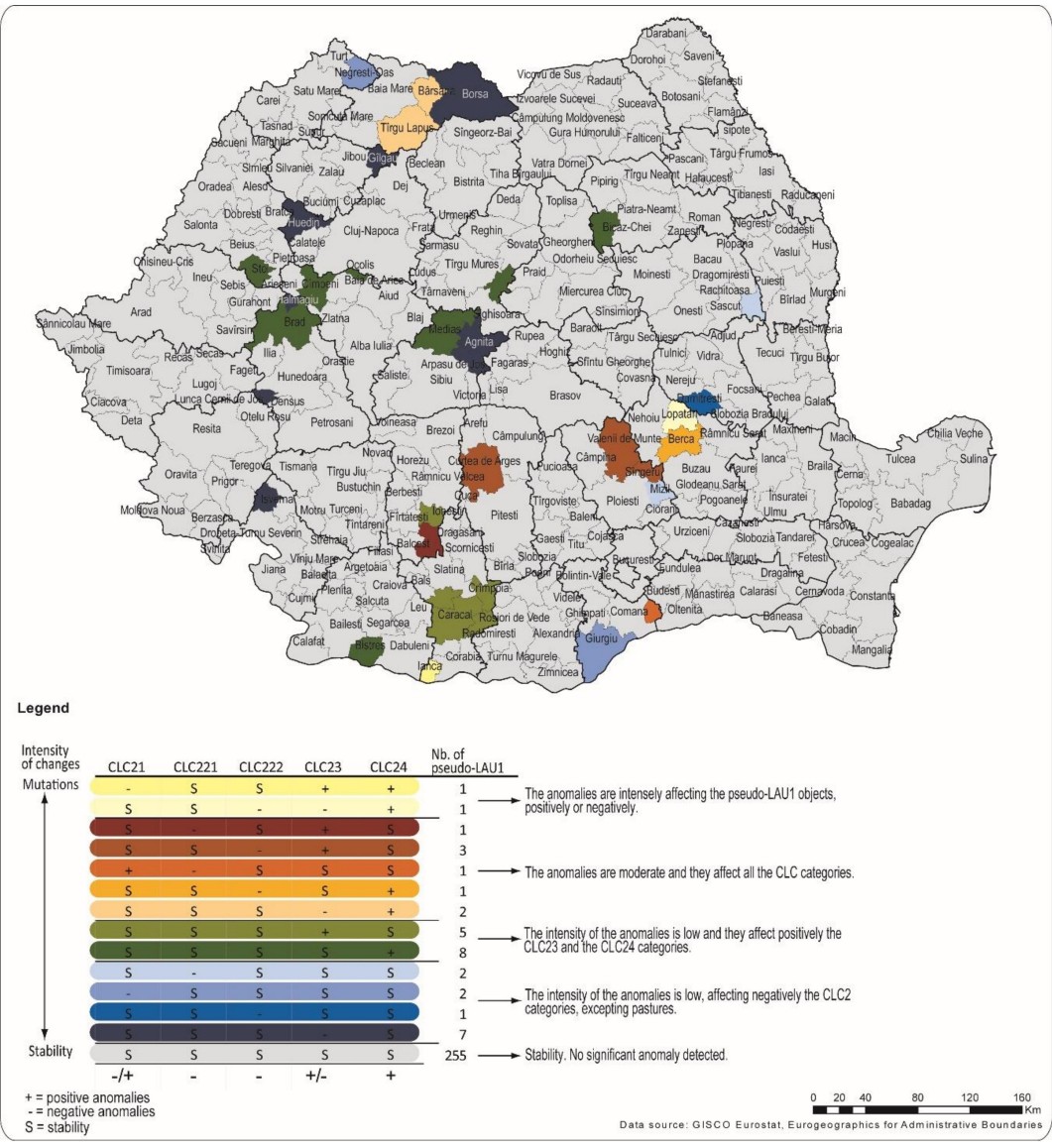

**Figure 3.** Changes in agricultural land-use patterns, an anomalies qualitative approach (2006–2018).

According to the data and to the typology, the most stable CLC classes are the arable land and the heterogeneous agricultural areas (CLC21 and CLC24). As spatial repartition, the dynamics are more likely to occur in the median longitudinal part of Romania, compared to the western border counties or the eastern periphery. It is not surprising the fact that a consistent amount of pseudo-LAU1 detected as anomalies for the agricultural dynamics are located in the Romanian mountain area, suggesting that the shifting patterns might also be correlated with altitude. Furthermore, no spectacular decline is visible in the proximity of the Romanian large cities (Bucharest, Timisoara, Cluj-Napoca and Iasi), making us consider that the inter-class changes (agricultural to artificial) are poorly targeted by the CLC datasets.

Addressing all the complexity of land-use transformations in Romania in one article is not a feasible option. We have considered that a proper way to better understand the new trends of land use and its anomalies relies on case studies. The choice for case studies related to the land-use transformations is a function of balancing opportunities to introduce a clearer view on the transformations against data access to possible validations of these changes with available data. We made an option for two case studies concerning the transformations of the agricultural use (arable land and vineyards) and one case study that focuses on the wet lands changes, in relation with both natural and society lead factors in the dynamic processes. The deforestation and afforestation phenomena were not included in our study for various reasons, such as: large surfaces of the mountainous forests were affected by strong winds and we cannot separate the natural phenomena from human influence in some areas; furthermore, sometimes the used deforestation methods were not "clear cuts" larger than 5 ha, but rather a selective cutting inside the forest and these are difficult to emphasize and they are visible only by applying a difference raster made from NDVI values on each reference year. These facts affect the final results and require supplementary information in order to study the forest area dynamics.

### 3.2. Case Study–Arable Land Transformations (CLC21)

The General Cadastre of Romania and Ministry of Agriculture and Rural Development from Romania [39,40] estimate that the total arable land in Romania is around 9.4 million ha, with small differences between 1990, 2007 and 2019. A total of 8.3 million ha of these are located in agricultural holdings. In accordance with these data, the CLC classification confirms that the area occupied by the arable land within Romania is around 8.15 million ha (1990, 2000), and it stabilizes between 8.73 million ha (2007) and 8.64 million ha (2012, 2018). The difference from the General Cadastre database is explained by the limitation imposed by the working scale proposed by the CLC methodology and the differences between different periods are caused by the quality (resolution) of the input data used for CLC 1990 and CLC 2000, which overestimated the heterogeneous agricultural areas, especially the classes 242 (complex cultivation patterns) and 243 (land principally occupied by agriculture), with significant areas of natural vegetation.

Although, through the CLC 2006 version, a consistent part of these differences was corrected, we consider that for the following versions, the methodology needs to be improved by deepening the degree of detail. In order to obtain more reliable results in accordance with the reality of the field, we also propose to use information resulting from field validation. For example, regarding the permanently irrigated land (code 212), the CLC data shows only a maximum of 2804 ha for 2018, while the National Agency for Land Improvement Agency [41,42] reported for the same year over 0.66 million ha, from which around 0.25 million ha was only in Braila county. Furthermore, the irrigated lands are relatively common in Ialomita, Dolj, Galati and Bacau.

Overall, the use of the CLC methodology is a good instrument for arable land-use/cover assessment within the limits of the methodological parameters, both in general spatial inventory to identify general trends as well as in highlighting different local specific patterns (Figure 4). For example, the abandonment of arable land in rural areas caused by the massive movement of local population to other countries or large cities (Maramures, Telorman, Giurgiu) can be easily and correctly highlighted.

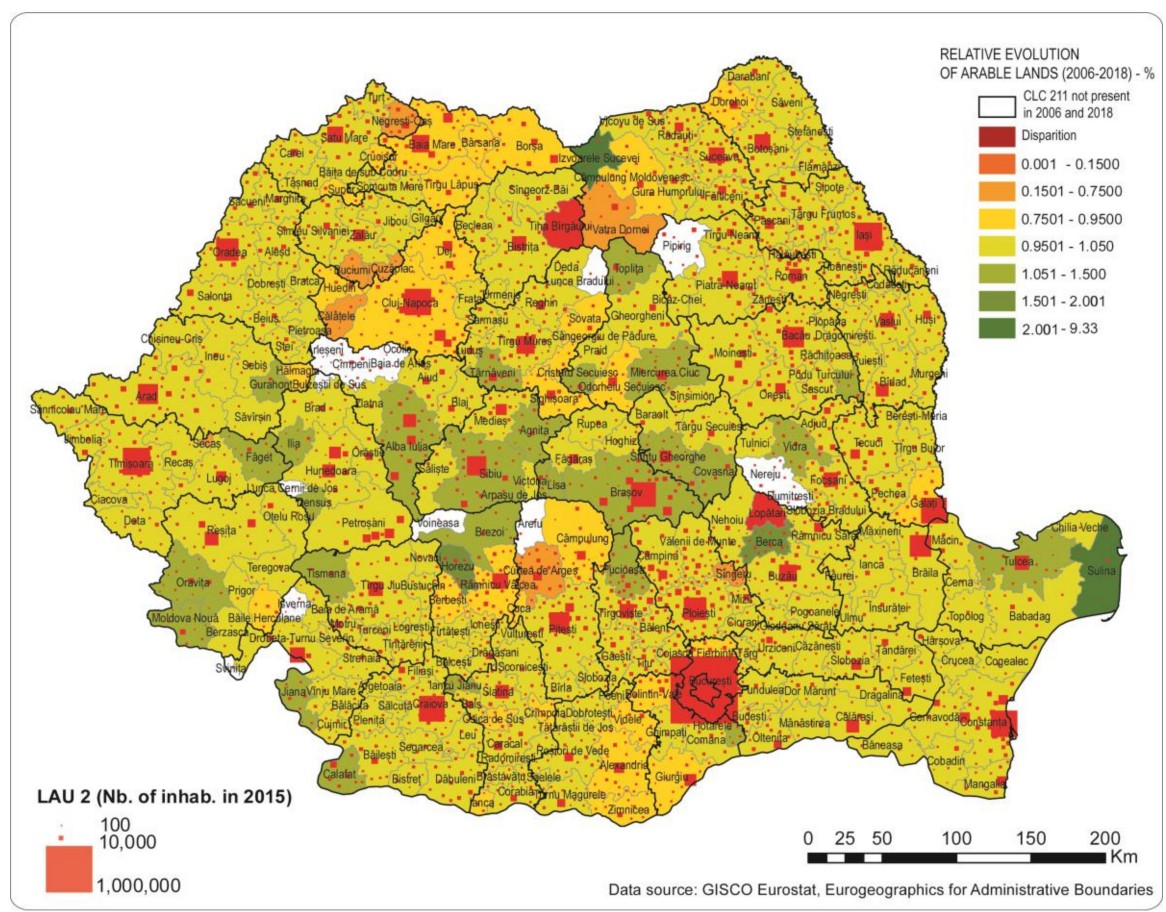

**Figure 4.** Evolution of arable lands in Romania between 2006 and 2018 according to CLC classification.

The consistent reduction of arable lands due to a permanent need of big cities for development is observed around Bucharest (Ilfov County), Cluj-Napoca, Iasi, etc. There are also examples of a substantial increase in arable land on account of the reduction of other land-use/land cover classes such as vineyards (PoduTurcului–Bacau). On the other hand, the influence of the financial infusion from the EU in the case of some traditional agricultural areas based on widespread field crops can be observed (Figure 5). This is the case in Tulcea, Sascut–Bacau, Iasi and Sfantu Gheorghe–Covasna. Unfortunately, the structural changes within the areas with a high percentage of arable land (pseudo-LAU1 of Timis, Arad, Slobozia, Ianca–Braila and Saveni–Botosani), affected by consistent internal transformation based on modern farming practices (land merging into large plots), are very difficult to detect.

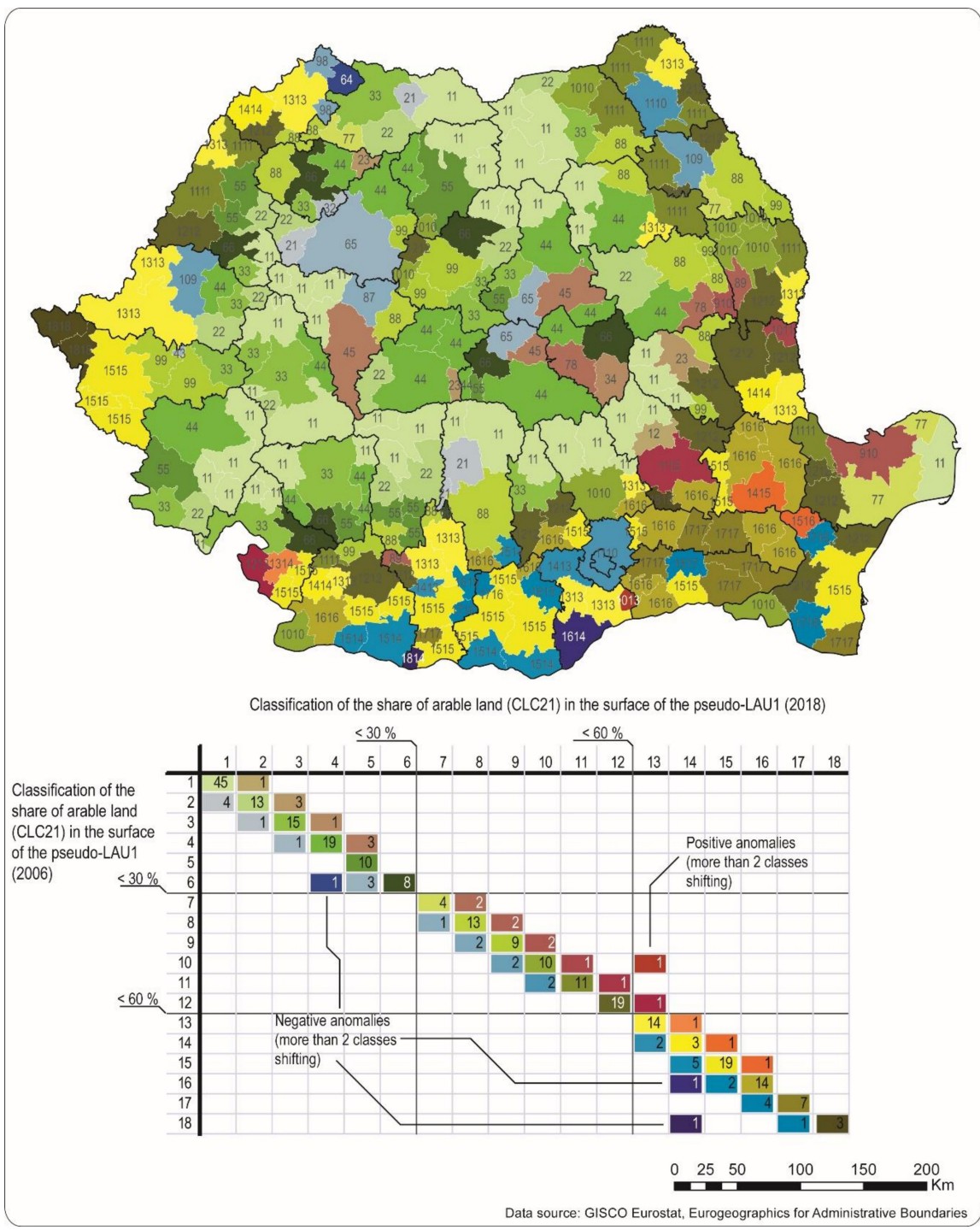

**Figure 5.** Evolution of arable land in Romania between 2006 and 2018 according to CLC classification and transition matrixes analysis.

### 3.3. Case Study–Vineyard Transformations (CLC221)

Analyzing the evolution of vineyards in Romania between 2006 and 2018, we note that this shift in climate favorability for vineyards is not directly observable in land use [43]. On the one hand, we can observe, for example, that in the north of the Moldova region, the vineyard surfaces are stable or even increasing (as in Botoșani pseudo-LAU1), a fact that can be considered an effect of the increased climate favorability for vines in these areas (Figure 6).

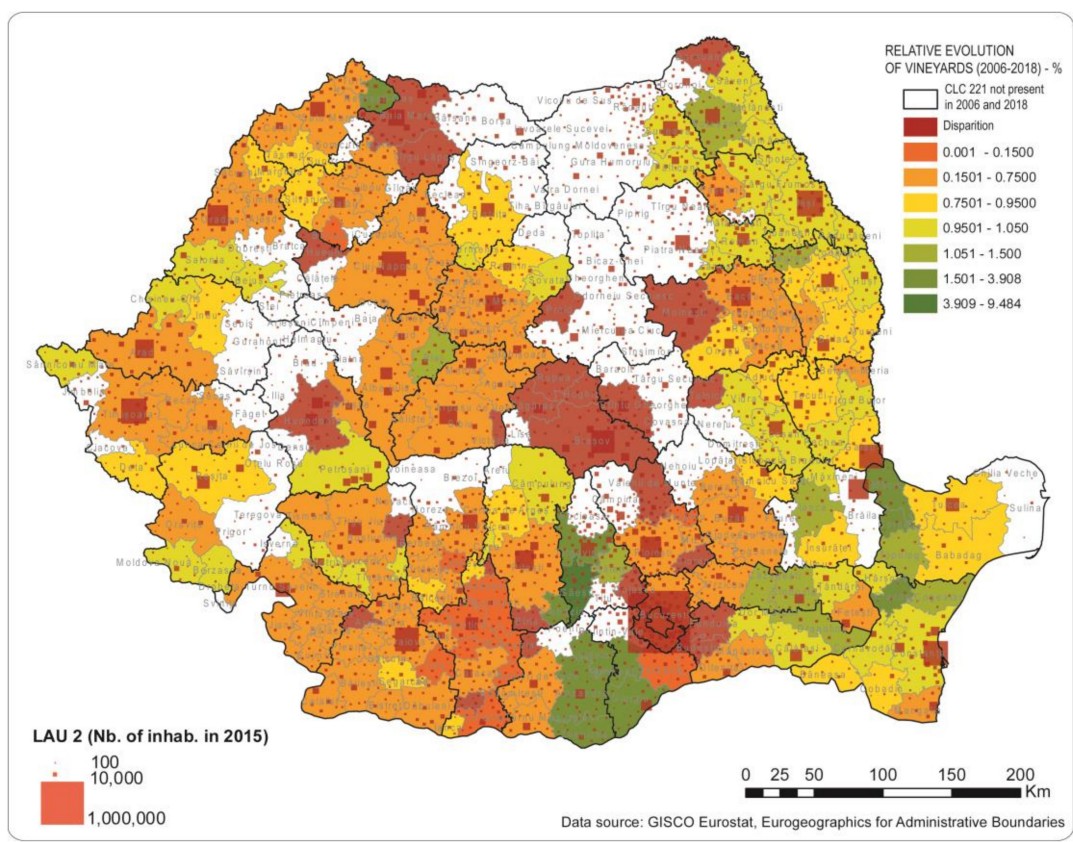

**Figure 6.** Evolution of vineyards in Romania between 2006 and 2018 according to CLC classification.

On the other hand, despite the decreased favorability expected for vines in the southern parts of Romania, some regions here (western Giurgiu, eastern Teleorman or Dâmbovița counties) reported an extension of vineyards between 2006 and 2018 [44]. Moreover, in this area we find only 3 Romanian pseudo-LAU1, indicating an important increase in the share of their territory occupied by vineyards. In this last case, it is obviously that the extension of vineyard surfaces represents the investors' voluntary choices to develop new vineyards, ignoring future climate conditions.

Overall, the evolution of vineyards is driven rather by the general trends of post-socialist transformations or by subjective and profit-maximization choices of agricultural specialization, regardless of the climate favorability.

In this approach, the prevailing declining trend observed in Romania for vineyard surfaces is explained mainly by the collapse of the exploitation developed during the period of planned agriculture before 1990. This is valid especially for extended regions in southwestern Romania, Prahova and Buzău counties or the south of Moldova. In some cases, the decrease in vineyard shares in the total surface is impressive at a pseudo-LAU1 level (Drăgășani-Vâlcea, Mizil-Prahova, Slobozia Bradului-Vrancea), despite the installation of new investors in the wine industry in this area, as in Drăgășani-Olt, in the last years. Actually, the new investors in the wine industry contributed to a stabilization of the vineyard shares in land use (Huși, Cotnari, Blaj or Vrancea NUTS3).

In other cases, the decrease of vineyard surfaces is not so pronounced due to the interests of local farmers for keeping or even developing small exploitations for self-consumption (Bistrița-Năsăud, Vaslui and Tulcea counties or in some pseudo-LAU1 from Bărăgan plain regions). In two pseudo-LAU1 especially (Negrești-Oaș and Negrești-Vaslui), this local incentive was big enough to lead to an important relative increase of vineyard surfaces, despite the limited climate favorability for vineyards in those regions. However, this interest does not dramatically change the share of vineyards within the corresponding pseudo-LAU1.

The vineyard dynamics in Romania works as an illustration of this methodological import from the theory of the economic convergence [45,46]. The diagonal of the matrix conserves data regarding the stagnation of the vineyard surfaces. The numbers above the diagonal indicate how many pseudo-LAU1 objects faced a positive evolution of the vineyard surfaces. The area below the diagonal of the matrix indicates how many pseudo-LAU1 registered an involution of the CLC221 surfaces. In this case, an equal interval classification was used, but other CLC categories might demand a different method. About 88% of the pseudo-LAU1 is placed on the diagonal of the matrix, indicating that the dynamics of CLC221 are limited (stability of surfaces). The decline of the category is present in 30 pseudo-LAU1 objects and generally involves a limited involution of this category, with three exceptions: one passage from class 5 to class 2 (Appendix A), one passage from class 6 to class 4 and two passages from class 3 to class 1 (Appendix A). This type of class shifting is the equivalent of more than 6% of CLC221 relative surface loss, at the scale of the pseudo-LAU1 objects. Taking into account the organization of the transition matrix for vineyard surfaces and its general stability, these four pseudo-LAU1 objects can be considered as anomalies in the evolution of CLC221 (Figure 7).

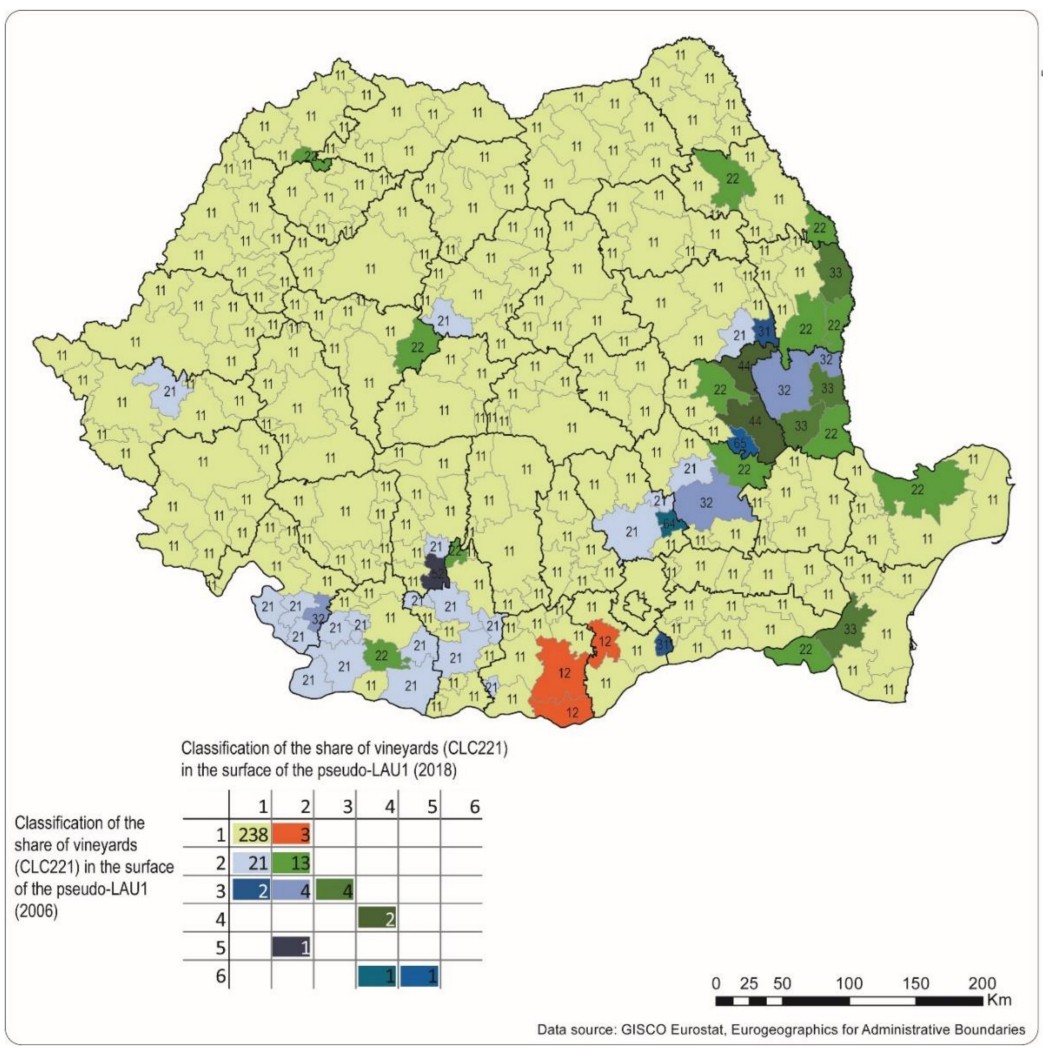

**Figure 7.** Evolution of vineyards in Romania between 2006 and 2018 according to Corine Land Cover classification and transition matrixes analysis.

## 3.4. Case Study–Wet Areas (CLC4)

The modifications of the surfaces included in the category of wetlands and those covered by water are induced by a series of anthropic and natural mechanisms. The aquatic surfaces take over and

best reproduce the natural effect of climate change and hydrological extremes. The correlation matrix between the surfaces covered with water in the period 2006–2018 shows a low correlation coefficient for this category ($r = 0.78$) as a reflection of the impact of the natural conditions on the aquatic units and their vulnerability in front of changes introduced by the anthropic factor.

The anthropic mechanisms are difficult to quantify and depend on the economic evolution and the climatic conditions specific to a country. In general, the changes made by collecting water volumes for various social and economic uses (water supply of the population and electricity production based on the volumes accumulated in the natural and anthropic lakes) [47,48] induce a series of changes of the surfaces covered by water at the level of large accumulations, but does not justify the important changes in the categories of use over a longer period of time.

In the period 1990–2018, the transition from CLC512 to CLC411 or even CLC231 was more evident at the level of 2012 (over 250 ha). For category CLC411, a decrease of 25% of the area initially occupied was observed between 2000 and 2006, compared to 1990. Most of these changes were registered in the low areas of the western (84%), center (64%), east (53%) and southeast (26%) parts of the country, while the lowest is in the Carpathian area (between 1 and 5%) (Figure 8).

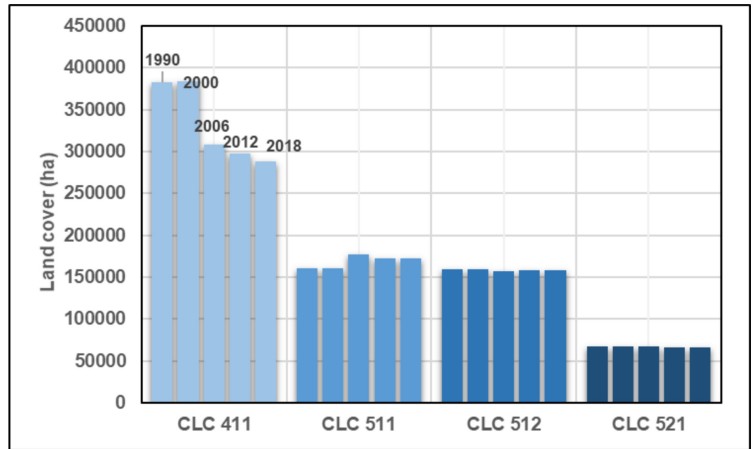

**Figure 8.** Changes in the areas occupied by categories CLC411, CLC511, CLC512 and 521 between 1990 and 2018.

A part of the decrease of the surfaces included in category 411 is found in CLC511 (approximately 22% of the area that is no longer in CLC411 at 2006 level is included in CLC511). The explanation lies in the way of reinterpreting the aquatic units as an effect of natural phenomena such as the large floods that occurred in Romania between 2005–2010 [49,50], with a maximum along the Danube valley in 2006 [51]. The temporary occupation with water of land surfaces during floods can create the false impression of including them in the category of watercourses [52].

The other part of the decrease of the areas included in CLC411 since 2006 is found at the level of CLC211, CLC231 and, less often, CLC243, probably as an effect of the economic evolution of the country under the impulse of inclusion in the European Union from 2007. For the period 2012–2018 the inclusion of important surfaces that initially belonged to CLC411 in the category of agricultural lands were also realized under the impulse of the natural droughts produced under the effect of the prolonged drought phenomena that affected the south and east of Romania in 2012 and 2015 [53]. The impact was reflected in the reduction of the areas framed in the category of wetlands and those covered with water, and the increase of those framed in the agricultural field only. This process can be easily classified in the category of anomalies in the evolution of land use, but it represents nothing more than a reflection of the natural hydrological cycles (with normal, drought and flood periods) over which anthropic interventions directed towards an economic valorization overlap. The largest decrease of the areas included in CLC411, CLC511 and CLC512 occurred between 2000 and 2006, with over 75% of these areas (40,000 ha) being found in the CLC211 (12,500 ha) and CLC 231 (25,000 ha) (Figure 9).

The mapping tools used to explore these changes are based on the transition matrixes for the CLC4 and CLC5 categories. The cartographic output indicates that the level of anomalies is low and it positively affects the pseudo-LAU1 of Maxineni (in NUTS3 Braila) and negatively affects the pseudo-LAU1 of Tulcea (in the NUTS3 Tulcea). All the other cases of class shifting concerns rather a limited amount of CLC transfer from wet areas and water bodies to other categories. All these changes tend to concentrate in the southeast region of Romania, mainly in the Danube Delta Natural Reserve (Figure 10). Further research is needed to better understand why, in a strictly protected area, wet surfaces (CLC4) are reducing in size.

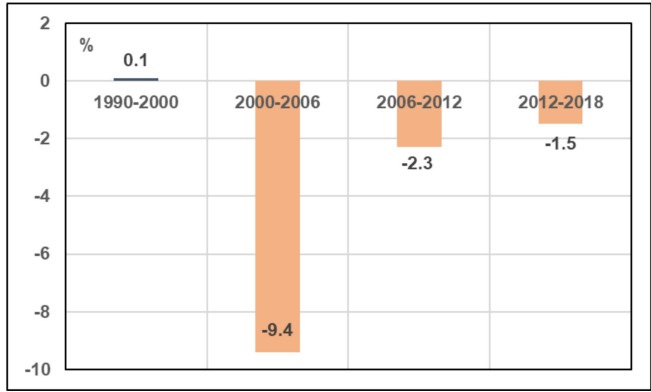

**Figure 9.** Increase/decrease of the areas occupied by categories CLC 411, 511 and 512 between 1990–2018.

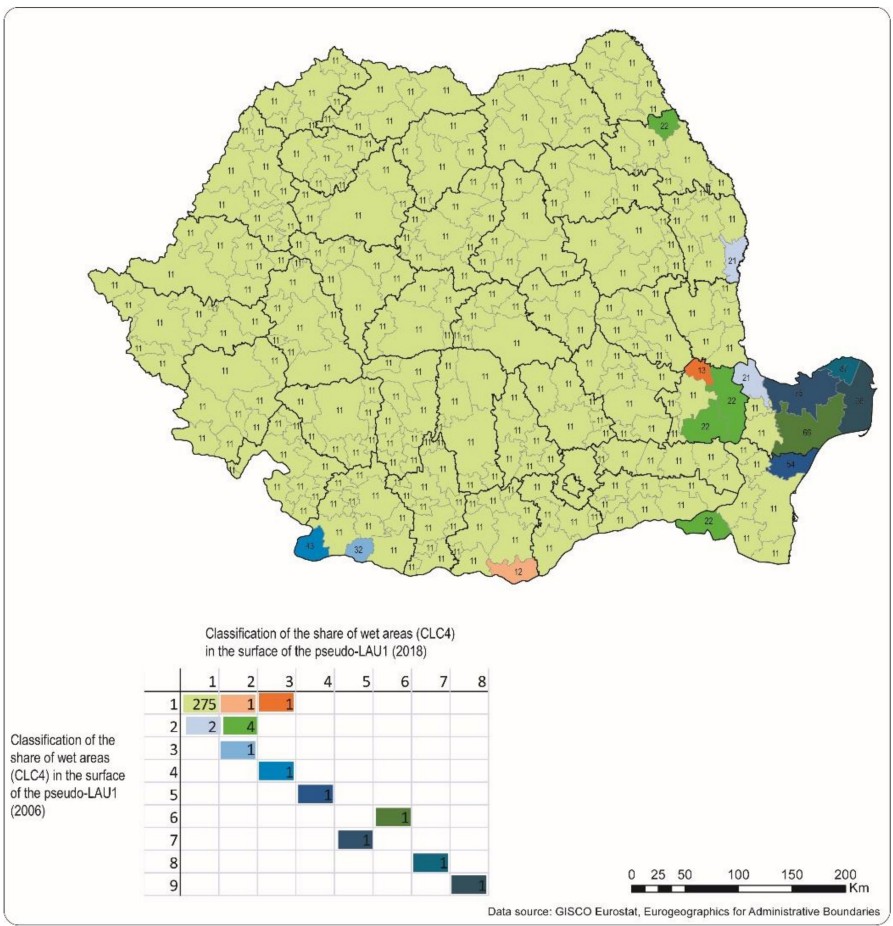

**Figure 10.** Output of transition matrix analysis for water bodies (CLC5) – low level of anomalies. Only differences of two classes can be considered anomalies (e.g., – 75 or 13).

The sequence of extreme hydrological events reflected in the way the land is covered with water is closely linked to the climatic changes that have occurred in the last decades on the Romanian territory [54–56], which can generate exceptional situations and errors of interpretation of the use of the land.

## 4. Discussion

Over the years, CLC evolved significantly in some aspects. For instance, the number of participating countries increased from 26 in 1986 to 39 in the present. The needed time to finalize the complete yearly dataset dropped from ten years to one year and a half. The satellite images used for the interpretation has a better time consistency and geometric accuracy of satellite data due to the change of the satellite type, from Landsat - 5 MSS TM in 1986 to Sentinel 2 in the present. Furthermore, the documentation is standardized and the results are free access for all users.

Some other things remained unchanged in the methodology during the years, such as minimum mapping unit 25 ha, minimum mapping width 100 m, change mapping boundary displacement should be minimum 100 m and all changes ≥ 5 ha are to be mapped.

Maps at finer scales are better for the study and modelling in detail, whereas maps at coarser scales are more suitable for the study and modelling of macro-structure and macro-changes [57].

The generalization process due to MMU and MMW variation ends with dominant classes prevailing over the rest of the categories [58]. On the other hand, when making use of fine-scale maps, we must consider that they are more likely to be affected by mapping mistakes [59]. Depending on the relative importance of these categories in explaining the pattern or dynamics of the classes of interest, one or other scales should also be selected [60]

We consider that a multi scalar dataset containing several scales will increase the number of applications connected to CLC. The increased quality of satellite images resulted in a better thematic accuracy but could not improve geometric features due to methodological limitations.

For our study we intended to use the land cover change files for the intervals 1990–2000, 2000–2006, 2006–2012 and 2012–2018. When we first checked the data files, we observed that some of the changes that should exist at the local level were missing from the files. We thought that it was a simple error and tried to compensate for the lack of information by using the difference from general land cover files from 1990–2000–2006–2012–2018 and obtaining other change files.

On the Copernicus Portal, the researchers were informed that due to differences in MMUs, the difference between two status layers will not equal the corresponding CLC-Changes layer. If you are interested in CLC changes between two neighbor surveys, always use the CLC-Change layer, enquiring CLC change databases is encouraged, and direct comparison of individual CLC layers is discouraged [61].

For analysis of land cover class total area on different dates, the CLC change layer is not practical: total gains and losses for each LC class should be calculated and added to the initial status layer. This process should be repeated for each CLC period (i.e., 1990–2000, 2000–2006, 2006–2012). In addition, this option combining information from a status and a change layer requires consideration of the different MMU (25 ha and 5 ha, respectively) [62].

Our study confirms that for Romania, when comparing the change files that resulted from our layer of difference with those downloaded from the EEA portal, we observe some problems. The values of the total changed area resulting from the difference are similar to those obtained from Land Cover Changes for the period 1990–2000 and 2012–2018. For the period 2000–2006 and 2006–2012 there are huge discrepancies between the declared changes and the "real changes". The real changes were 16 times greater than the declared one between 2000 and 2016 and 8 times bigger in 2006–2012. It is clear that some corrections were made in the land cover files from 2006 and 2012 and they were not recorded on the changes (Figure 11).

Our research highlighted which category suffered more corrections in these two years. From our data, the results show that the CLC 211 has increased dramatically in 2006, probably by reducing the

CLC243 and CLC242 category, and we think that this was the result of better satellite data. We also note significant decreases for CLC112, 221, 324 and 411. In 2012 there was an increase of CLC231, 324 and 112 and a decrease of CLC211, 221 and 222.

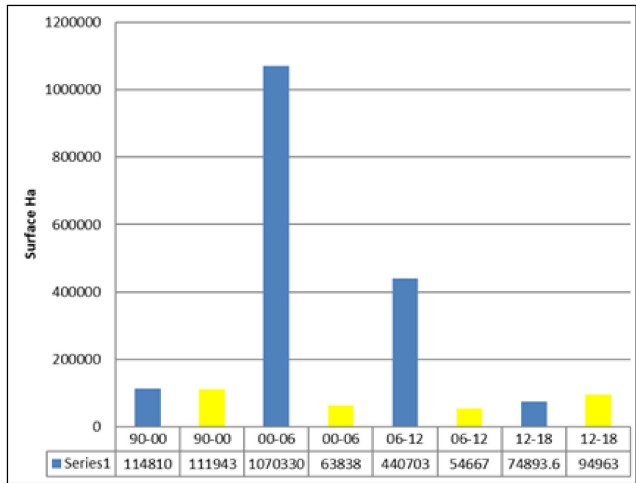

**Figure 11.** Statistical assessment of land-use changes (1990–2018). Total surface change resulted from layer difference (blue) or from the change files (yellow).

Some categories, that registered a significant raise of total surfaces in 2006, showed an opposite trend in 2012, such as CLC211, 231 and 324. If we analyze the total change surface per year and category, we see that the most dynamic classes are those related to deforestation and afforestation (CLC324, 312, 311 and 313, Figure 12).

Another dynamic group is the agriculture group (CLC211, 221 222, 242 and 243). The changes in this group are similar to those from the difference files, but the magnitude is much smaller. The surfaces of the artificial classes are also increasing, instead of decreasing, as presented in the previous graph. The increase of the artificial surfaces is a logical process if one takes into account the major trends of urban sprawl detectable each year after 2006, which accelerate after 2016 [63]. Both independent real estate market analysts and the Romanian NSI (National Statistical Institute) offer data that corroborates this statement. In some cases, the artificial surfaces are transformed only from a functional point of view, within the same CLC category (some industrial brownfields are upgraded to commercial sites). In other cases, located remotely to the city centers, these transformations are affecting the agricultural areas, which are converted to urban fabric. If we merge in the same graph the changes from differences and the official changes, we observe that the files from 1990–2000 and 2012–2018 are comparable and, even if there are slightly different values for the surfaces, the dominance of some types of changes is confirmed by both comparative methods. Probably the 1990–2000 files have a certain consistency due to the fact that it was necessary for a long period of time to create that database, and also there was a large amount of working time allocated to the harmonization. The 2012–2018 period coherence is probably explained by the fact that the satellite data are better and the historical errors were corrected in 2006 and 2012. The qualitative approach of anomaly detection by transition matrixes at the pseudo-LAU1 scale largely confirms the fact that the land-cover dynamics demand a multi-scalar and multi-methodological approach, for a better understanding. Long-term studies comparing early layers and recent layers may find discrepancies related to methodological issues rather than real land cover changes [64].

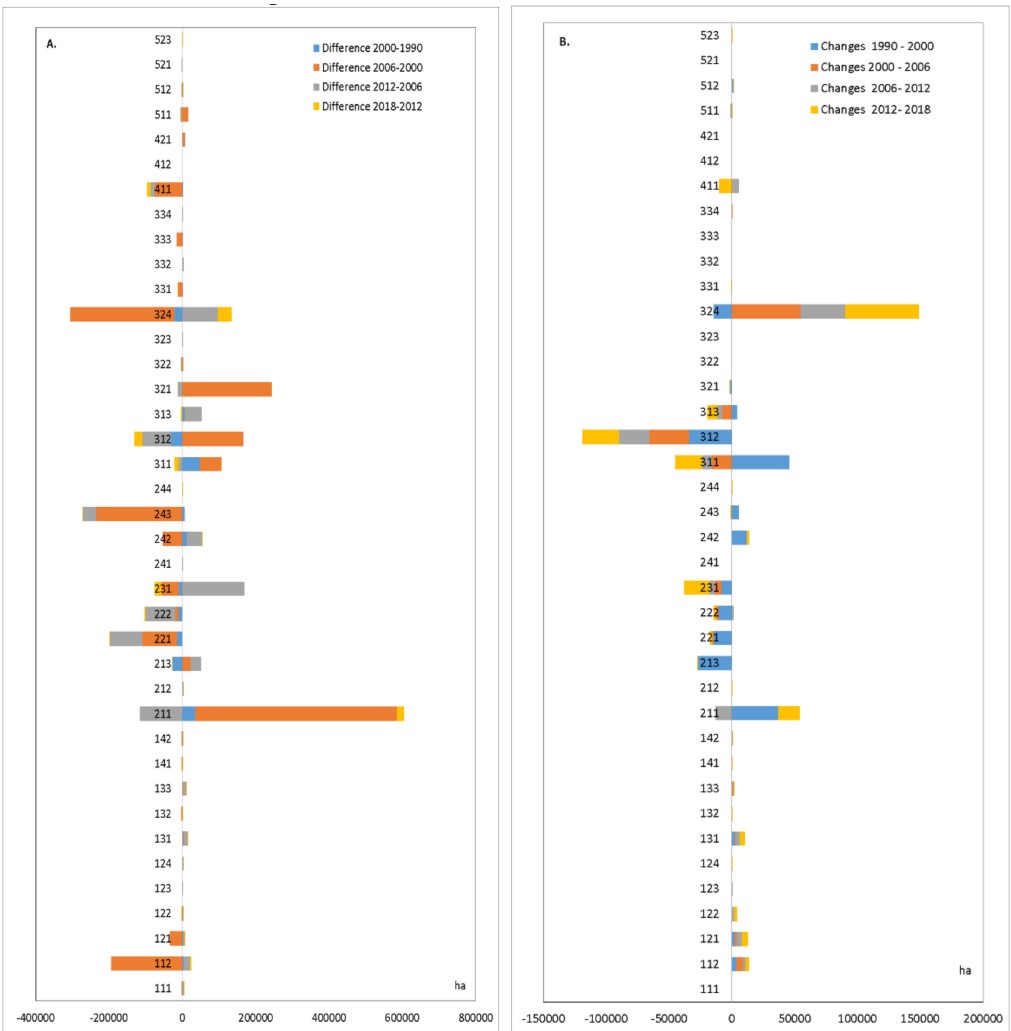

**Figure 12.** Land cover changes resulting from the difference of base layer classification (1990–2018) (**A**) and land cover changes (1990–2018). Official data from the Copernicus Portal, surface (ha) (**B**).

For the 2000–2006 and 2006–2012 period, the changes resulted from the differences and those downloaded from the Copernicus have the greatest discrepancies. It is not just a quantitative difference but also a significant qualitative one. These differences must be generated by large-scale corrections that were made on a single year. Probably, the errors that have been corrected should have also been translated on the layers of the previous years or, they should have been translated to change files. Otherwise, the comparison is almost impossible, in the case of Romania. The magnitude of differences is huge—over 500,000 ha of arable land appear on the 2006 land cover layer. Furthermore, the spatial distribution of these corrections is quite generalized, as we can see in Figure 13. In an expert opinion approach, our analysis emphasizes the fact that the mutations between CLC categories can be expected to occur much more in mountain areas or in the proximity of large urban zones.

Using the land cover change files for the entire transition period demands precautions for Romania, because they show different results compared to reality, both from the thematic and the quantitative perspective. Using the difference layers of the land cover for each interval is also not advised, due to huge differences between them, so we have decided to use the unique difference layer between the 2006 land cover layer (before Romania accession to EU) and that from 2018, in order to investigate the difference between these two milestones in Romanian history.

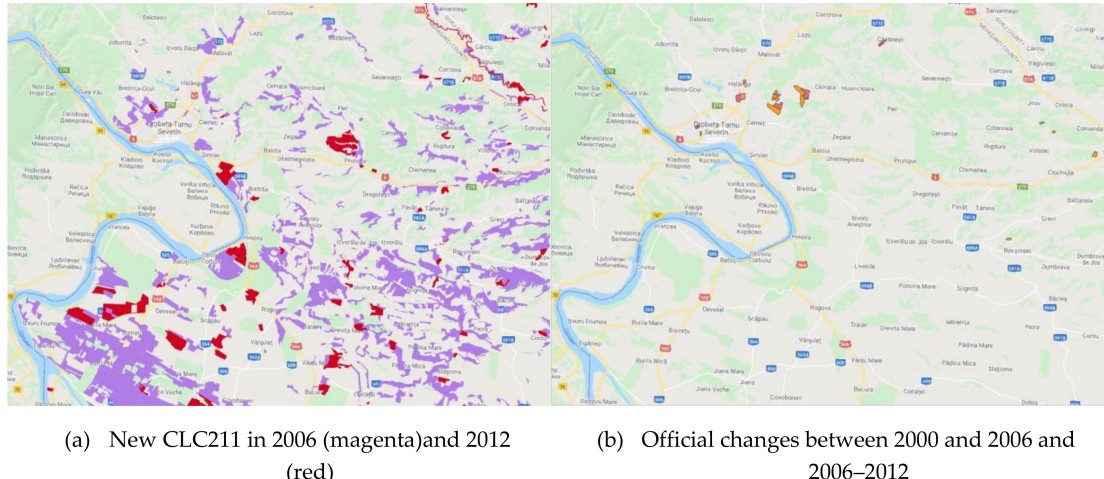

(a) New CLC211 in 2006 (magenta)and 2012 (red)

(b) Official changes between 2000 and 2006 and 2006–2012

**Figure 13.** (**a**,**b**)Technical GIS tools applied for a better understanding of land-use dynamics.

From a qualitative change assessment point of view, different approaches might be taken into account. The first one will emphasize potential technical changes related to the data quality check. The second one will rely on cross-checking of the data process, involving local or regional alternative data. Such is the case of the industrial park endowment or for the new agro-industrial facilities located in the rural areas of Romania. Both of them are strongly interrogating the dynamics of the artificial land-use changes, as was delineated by the CLC datasets in 2006 and 2018 (Figure 14).

However, as long as the MMU in CLC still remains at its value of 25 ha, due to time consistency challenges one will hardly recognize any mutation in these land-use categories. Moreover, for a better future validation of the CLC land cover and changes during time, the scientific modeling derived from an interdisciplinary approach should be considered a priority. For example, planners, remote sensing analysts and geographers co-working will better address the issue of industrial facility location in Romania, as a function of multiple decision-taking processes, e.g., proximity to markets, accessibility or speculative land opportunities. Basically, some of these mentioned location opportunities will lead to a progressive change of land-use patterns in the peripheral areas of a selected set of Romanian cities.

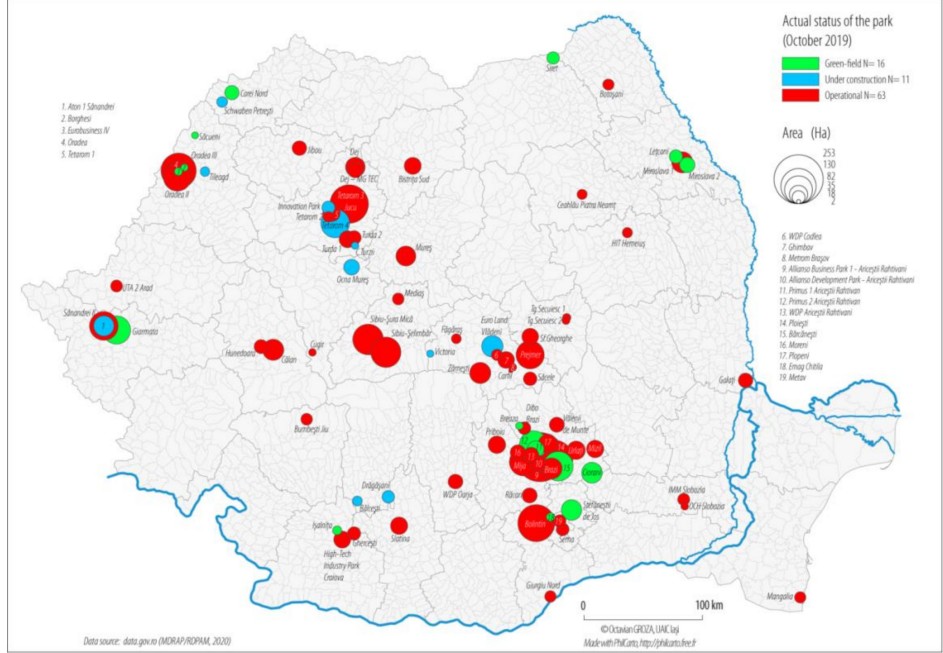

**Figure 14.** Extension of industrial parks in Romania (cumulative data until October 2019).

This kind of transformation is already at work but it is hard to be captured by a MMU of 25 ha in the CLC project, when the rhythm of accumulation of new land uses and dynamics is not rapid enough during this time. The official sources at a local level, especially in the case of the industrial park land-use accumulation, cope better with the reality than the CLC does. This fact is not a criticism for the CLC datasets; we are just underlining the potential of alternative data source exploitation, combined with adjusted new functional geometries, such as the pseudo-LAU1 one.

## 5. Conclusions

The CLC program is a very useful tool, but it should be improved in the future, considering that since its inception many things have improved significantly, such as satellite image quality and frequency over time, their coverage of the target area and knowledge of the researchers involved. We consider that a multiscale approach of the land cover change issues in Europe would be more appropriate for the present necessity of the society, so we think that it is necessary to 1) maintain the current scale 1:100,000, in the construction of the CLC layers, in order to be able to compare new data with the layers from the previous years; and 2) add a supplementary layer 1:25,000 in order to achieve more refined results in the future, which will be correlated with the new imaging possibilities and with the additional information available in 2020. On this new layer an update of methodology and of the CLC nomenclature is recommended.

Our study, in Romania, at intermediate scales (pseudo-LAU1 scale) proves that the anomalies of land use can be detected using transition matrixes and a qualitative mapping approach. In the Romanian case, the interpretation of the transformations concerning the arable land (CLC21) put an emphasis on the role played by the major economic actors investing in agriculture. The case of vineyards (CLC221) is more complicated because the transformations might have a double lead force—climate change and new economic specializations in some particular regions. The wet areas and water bodies case study (CLC4 and CLC5) show that some areas of natural land cover are subject to changes, even if they concern areas with sound management and protection, such as the Danube Delta Nature Reserve. We consider that many of the CLC change anomalies might be explained by targeted analyses and case studies focusing on specific areas or land-use categories, our final recommendation. This kind of analysis can be extrapolated to other regions. The main limitative aspect of our research, excluding the micro-data consistency regarding the land use in time, is related to the construction of the alternative pseudo-LAU1 geometry. Other methods of LAU2 aggregation (hierarchical clustering with spatial constraint) will produce different frameworks that need to be explored. The method we used for the delineation of land use anomalies is based on transition matrixes, without encompassing the probabilities of change from one chronological milestone to another.

The added value of this method relies both on a methodological transfer from the field of economic geography (theory of convergence) and on the more friendly interpretations for policymakers.

The case studies proved that the combination of natural and socioeconomic factors creates changes that are sometimes counterintuitive, especially when they are reported to the basic assumptions that scientist make regarding the dynamics of land cover.

**Author Contributions:** Conceptualization, A.U. and O.G.; methodology, A.R. and A.U.; software, C.C.S. and A.R.; validation, L.N., L.S., I.M. and O.M.S.; formal analysis, O.G. and A.U.; investigation, L.N., L.S., I.M. and O.M.S.; resources, A.R. and C.C.S; data curation, A.U. and C.C.S.; writing—original draft preparation, A.R. and I.M.; writing—review and editing, A.U., I.M. and O.M.S.; visualization, A.R. and I.M.; supervision, O.G. All authors have read and agreed to the published version of the manuscript and consider that have contributed equally to the present article.

**Funding:** This project is funded by the Ministry of Research and Innovation within Program 1— Development of the national RD system, Subprogram 1.2—Institutional Performance—RDI excellence funding projects, Contract no. 34PFE/19.10.2018.

**Acknowledgments:** This project received technical support from Department of Geography, Faculty of Geography and Geology,"Alexandru Ioan Cuza" University of Iasi, Romania who offered us full access to the remote sensing and GIS laboratories.

**Conflicts of Interest:** The authors declare no conflict of interest.

**Appendix A**

**Legend of Corine Land Cover**

**Class 1. Artificial surfaces** include: *1.1. Urban fabric (1.1.1. Continuous urban fabric, 1.1.2. Discontinuous urban fabric); 1.2. Industrial, commercial and transport units (1.2.1. Industrial and commercial units, 1.2.2. Road and rail networks and associated land, 1.2.3. Port areas, 1.2.4. Airports); 1.3. Mine, dump and construction sites (1.3.1. Mineral extraction sites, 1.3.2. Dump sites, 1.3.3. Construction sites); 1.4. Artificial non-agricultural vegetated areas (1.4.1. Green urban areas, 1.4.2. Sport and leisure facilities)*

**Class 2. Agricultural areas** include: *2.1. Arable land (2.1.1. Non-irrigated arable land, 2.1.2. Permanently irrigated land, 2.1.3. Rice fields); 2.2. Permanent crops (2.2.1. Vineyards, 2.2.2. Fruit trees and berry plantations, 2.2.3. Olive groves); 2.3. Pastures (2.3.1. Pastures), 2.4. Heterogeneous agricultural areas (2.4.1. Annual crops associated with permanent crops, 2.4.2. Complex cultivation patterns, 2.4.3. Land principally occupied by agriculture, with significant areas of natural vegetation, 2.4.4. Agro-forestry areas)*

**Class 3. Forests and semi-natural areas** include: *3.1. Forests (3.1.1. Broad-leaved forest, 3.1.2. Coniferous forest. 3.1.3. Mixed forest); 3.2. Shrub and/or herbaceous vegetation association (3.2.1. Natural grassland, 3.2.2. Moors and heathland, 3.2.3. Sclerophyllous vegetation, 3.2.4. Transitional woodland shrub); 3.3. Open spaces with little or no vegetation (3.3.1. Beaches, dunes, and sand plains, 3.3.2. Bare rock, 3.3.3. Sparsely vegetated areas, 3.3.4. Burnt areas, 3.3.5. Glaciers and perpetual snow)*

**Class 4. Wetlands** include: *4.1. Inland wetlands (4.1.1. Inland marshes, 4.1.2. Peatbogs); 4.2. Coastal wetlands (4.2.1. Salt marshes, 4.2.2. Salines, 4.2.3. Intertidal flats)*

**Class 5. Water bodies** include: *5.1. Inland waters (5.1.1. Water courses, 5.1.2. Water bodies); 5.2. Marine waters (5.2.1. Coastal lagoons, 5.2.2. Estuaries, 5.2.3. Sea and ocean)*

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
