# Peer review of "Structural Changes in the Romanian Economy Reflected through Corine Land Cover Datasets"

_remotesensing, doi:10.3390/rs12081323_

Round 1
Reviewer 1 Report
Manuscript presents an interesting use of the Corine Land Cover database to investigate land use changes during adjustments of Romania to free market economy. Results have great potential and can be interest to the readers.
However manuscript is too long. It does not have proper structure and content of some chapters needs improvement. Thus I suggest several general and detailed changes that should clarify parts of manuscript.
Abstract
Reduce first part of abstract to max. 2-3 sentences, lines: 12-23
Add one sentence about methods and type and years of dataset used
Clarify results of your study, lines 28-40:
Remove sentence without meaning, lines 32-33: “Maybe the most interesting changes occurred in the case of the agricultural polygons.”
Avoid phrase in line 28: “For example,….”
Introduction
This chapter must be significantly modified and should contain following paragraphs:
Introduce the topic of your paper – its worldwide/regional relevance
What is known about topic – up to date studies, cite broadly worldwide
What is not known - clear description of problem
Specific aims - research questions and/or hypotheses at the end of introductory chapter
It is not clear what is a general aim of the study – Methodological? Analysis of land use changes? Both?
I suggest one general aim and/or hypothesis with 2-3 specific goals. There are several sentences in the manuscript indicating possible aims:
Page 1, lines 24-25: Our main intention for this study is to observe how these economic readjustments can be assessed and measured using the Corine Land Cover datasets over time.
Page 2, lines 45-47: This article investigates potential ways in which the CLC dataset can be used for more in depth analysis of land-use and land cover transformations, at an intermediate scale, appropriate for policy design and decision process in Romania.
Page 9, lines 349-350: Our initial hypothesis and research strategy tried to connect economic and social changes with potential mutations in the land use patterns, at local scale.
Page 24, lines 727-733: The syntheses of these discussions clearly gravitate around the main question: how to assess land use changes in Romania? A technical GIS approach that investigates them at local scale, obtaining cumulative micro-data? A qualitative classification based on transition matrixes applied on an alternative functional geometry (pseudo-LAU1)?
Explain abbreviations in places where they appear for the first time: NUTS and LAU
Page 2, lines: 70-81: This paragraph should be in Methods
Materials and Methods
The whole chapter should be more consistent and reduced at least by 30%
Page 3, lines 116-117 and in the whole manuscript: Explain land use categories that you use for the first time CLC112, CLC111 etc.
Page 5, lines 212-225: Paragraph is rather part of Results and/or Discussion.
Page 7, lines 256-260: Research paper has usually well defined structure and it is not necessary to present again its organization.
Page 7, lines 262-263: It is obvious that you could not analyse all CLC land use types, but it is not obvious why you selected arable land, vineyards and wet areas for detailed analysis. Explain it in this chapter.
Results
In results we avoid discussion with literature and these parts should be removed and/or incorporated to other chapters.
Pages 7-9, lines 265-336: Chapter 3.1. should be transferred to the Materials and Methods and can be entitled as subchapter 2.1.: “The main economic changes and land use transformation in Romania after collapse of communism”. This new subchapter should be shortened and some parts can be used in the Introduction
The existing text from Materials and Methods can be entitled as second subchapter 2.2: “Datasets and methods”
Page 11 lines 403-415: can be removed and/or shortened and/or transferred to the Introduction – it repeats some previous information that is not related to results.
Page 11, line 416: remove phrase ”Generally speaking..”
Page 13, lines 460-469: the first paragraph can be transferred to the Discussion.
Page 24, lines 727-733: Aim can be prepared on the basis of these sentences. Discussion chapter should answer on questions from aims. Thus it should not finish with questions.
Discussion
In the Discussion follow general and specific aims of your study!
Conclusions
Chapter needs strong modification. Some paragraphs/sentences seem more relevant for Discussion. Conclusions should be short and consistent with max. 2-3 paragraphs including:
- main conclusion
- implications for the research field
- potential future directions
Avoid descriptive sentences such as:
Page 24, lines 735-737: This section of the article is organized as a set of recommendations, illustrating the perspective of a heterogeneous group of researchers (remote sensing, GIS, human geography and classical physical geography) and their expert opinion.
Page 25, lines 769-771: The relevant key findings of our research will be exposed in the final part of this section. They include: specifications regarding the limitations of our methodological approach, a possible continuation of our study in further research and some basic conclusions.
Author Response
Manuscript presents an interesting use of the Corine Land Cover database to investigate land use changes during adjustments of Romania to free market economy. Results have great potential and can be interest to the readers.
However manuscript is too long. It does not have proper structure and content of some chapters needs improvement. Thus I suggest several general and detailed changes that should clarify parts of manuscript.
Thank you for your clearly observation. They helped us a lot to correct and to restructure the whole work.
Abstract
Reduce first part of abstract to max. 2-3 sentences, lines: 12-23
Add one sentence about methods and type and years of dataset used
Clarify results of your study, lines 28-40:
Remove sentence without meaning, lines 32-33: “Maybe the most interesting changes occurred in the case of the agricultural polygons.”
Avoid phrase in line 28: “For example,….”
We made important modification of the abstract and deleted the sentence without meaning. Now the abstract are significantly reduced
During the last 30 years, the Romanian economy faced different challenges due to structural readjustments, crisis overcoming and globalization. The share of the primary and secondary sectors in the Gross Domestic Product strongly decreased, while the services took-off. The main objective for this study is to observe how these economic readjustments can be assessed and measured using the Corine Land Cover datasets from 1990, 2000, 2006, 2012 and 2018 (with special observation on the range 2006 and 2018 after Romania was included in European Union). Despite some of the methodological limitations (like the minimum surface change), the Corine Land Cover turned out to be a powerful tool and it allowed us to detect an intense correlation between the socio-economic and the structural trends in land use, in specific spatial contexts. The artificial surfaces are constantly increasing and this trend is rather visible as a distance function to the major Romanian cities. The most interesting changes occurred in the case of the agricultural polygons. The main trend emphasized by our analysis regards the redeployment of large farms in areas of agronomic and environmental territorial optimum. Such is the case for vineyards (after a decline during the 2000-2006) and for annual cultures. All these changes in the land use patterns are too complex to be encompassed by a single methodology, that’s why we used different tools, ranging from spatial analysis to geo-economic modeling, in order to detect how the Corine Land Cover datasets might be used for a better understanding of the Romanian economic readjustments.
Introduction
This chapter must be significantly modified and should contain following paragraphs:
Introduce the topic of your paper – its worldwide/regional relevance
In this paper we investigate the land cover and land changes in Romania, using an appropriate scale of analysis – intermediate between the main administrative frame and the local one. The period we emphasized in our research covers mainly the 2006-2018 time intervals. The methodology we developed is applied at a national scale, but it can be replicated for other states too.
What is known about topic – up to date studies, cite broadly worldwide
After 1990, during the post-communist period, Romania passed through complex social, economic and political transformations which lead to radical background changes within some important fields such as land property and agricultural land exploitation. Some of these transformations were investigated at global [1], European [2] and national level using Corine Land Cover (CLC) datasets [3, 4, 5]. The CLC project largely fulfills the expectations of the researchers and other users [6], having two main advantages: it provides a seamless geometry usable for macro-regional studies and an internal classification of the land use/cover categories that allows time traceability of changes, for a reasonable period (1990-2018). The facility of tracking the changes in the use of the land has allowed a multitude of applications and correlations between CLC datasets and different parameters [7, 8, 9]. There is an intense use of the CLC datasets for local or regional studies; however, the intermediate scales are systematically neglected by other researches [10, 11, 12, 13]. That’s why we consider that the proper scale of CLC use for policy design should be an intermediate one, an administrative geometry that fits the needs of analysis somewhere between the local level and the NUTS3 delineation [14].
What is not known - clear description of problem
The use of CLC at local administrative unit (LAU2) is largely unadvised [15], as the degree of land use generalization might interfere with the local patterns. In this case, numerous policy makers will eventually use CLC just as an informative base of interpretation and not as an analytical frame. At NUTS3 scale, the CLC dataset overlaps national or regional statistical information that, in some conditions, creates data redundancy or impossible comparisons (mainly in the case of the artificial surfaces) [16].
In this paper we investigate the land cover and land changes in Romania, using an appropriate scale of analysis – an intermediate one between the main administrative frame and the local scale. The period we emphasized in our research covers mainly the 2006-2018 time intervals. The methodology we developed is applied at a national scale, but it can be replicated for other states too.
The problem of the intermediate scale of analysis is mainly related to its construction. If the studies at local scale generally involve a limited set of beneficiaries (policy makers), an intermediate scale would mean to aggregate administrative polygons belonging to a larger set of decision takers. One of the intentions of our study was to develop a method of administrative polygons aggregation that can be intersected with the CLC datasets. This intermediate scale of analysis was labeled pseudo-LAU1, where LAU1 stands for local administrative units of first rank. Using spatial analysis techniques and potential accessibility functions [17], was created an operational pseudo-LAU1 geometry that was used in order to collect the land-use and land-cover information provided in the CLC vector database (2006-2018). The building blocks of this operational geometry are the Romanian local administrative units known as LAU2.
Specific aims - research questions and/or hypotheses at the end of introductory chapter
Considering the ones described above the principal objectives of this article is to investigates potential ways in which the CLC dataset can be used for more in depth analysis of land-use and land cover transformations, at an intermediate scale, appropriate for policy design and decision process in Romania.
It is not clear what is a general aim of the study – Methodological? Analysis of land use changes? Both?
The general aim of this study is to analyses some methodological aspects of using CLC regarding with land use changes in Romania
I suggest one general aim and/or hypothesis with 2-3 specific goals. There are several sentences in the manuscript indicating possible aims:
Page 1, lines 24-25: Our main intention for this study is to observe how these economic readjustments can be assessed and measured using the Corine Land Cover datasets over time.
Page 2, lines 45-47: This article investigates potential ways in which the CLC dataset can be used for more in depth analysis of land-use and land cover transformations, at an intermediate scale, appropriate for policy design and decision process in Romania.
Page 9, lines 349-350: Our initial hypothesis and research strategy tried to connect economic and social changes with potential mutations in the land use patterns, at local scale.
Page 24, lines 727-733: The syntheses of these discussions clearly gravitate around the main question: how to assess land use changes in Romania? A technical GIS approach that investigates them at local scale, obtaining cumulative micro-data? A qualitative classification based on transition matrixes applied on an alternative functional geometry (pseudo-LAU1)?
Thank you for this clearly observation. we chose to write at the end of introduction part that: This article investigates potential ways in which the CLC dataset can be used for more in depth analysis of land-use and land cover transformations, at an intermediate scale, appropriate for policy design and decision process in Romania.
Explain abbreviations in places where they appear for the first time: NUTS and LAU
All the abbreviations where described after they appear for the first time
Page 2, lines: 70-81: This paragraph should be in Methods
We moved this paragraph to the methods part.
Materials and Methods
The whole chapter should be more consistent and reduced at least by 30%
All the chapter was restructured according to your observations.
Page 3, lines 116-117 and in the whole manuscript: Explain land use categories that you use for the first time CLC112, CLC111 etc.
We described in the text what means CLC112, CLC111 etc.
Page 5, lines 212-225: Paragraph is rather part of Results and/or Discussion.
We moved this paragraph to the discussion part
Page 7, lines 256-260: Research paper has usually well defined structure and it is not necessary to present again its organization.
We deleted this part from reviewed manuscript
Page 7, lines 262-263: It is obvious that you could not analyse all CLC land use types, but it is not obvious why you selected arable land, vineyards and wet areas for detailed analysis. Explain it in this chapter.
Results
In results we avoid discussion with literature and these parts should be removed and/or incorporated to other chapters.
Pages 7-9, lines 265-336: Chapter 3.1. should be transferred to the Materials and Methods and can be entitled as subchapter 2.1.: “The main economic changes and land use transformation in Romania after collapse of communism”. This new subchapter should be shortened and some parts can be used in the Introduction
The existing text from Materials and Methods can be entitled as second subchapter 2.2: “Datasets and methods”
We made all the modifications suggested.
Page 11 lines 403-415: can be removed and/or shortened and/or transferred to the Introduction – it repeats some previous information that is not related to results.
Some part of this text we moved to the introduction.
Page 11, line 416: remove phrase ”Generally speaking..”
We removed the phrase.
Page 13, lines 460-469: the first paragraph can be transferred to the Discussion.
Parts of these lines were moved to the discussion chapter.
Page 24, lines 727-733: Aim can be prepared on the basis of these sentences. Discussion chapter should answer on questions from aims. Thus it should not finish with questions.
Thank you for this suggestion we moved part of this text to the introduction.
Discussion
In the Discussion follow general and specific aims of your study!
The discussion part after all the modification made follow now the general and specific aims of our study.
Conclusions
Chapter needs strong modification. Some paragraphs/sentences seem more relevant for Discussion. Conclusions should be short and consistent with max. 2-3 paragraphs including:
- main conclusion
- implications for the research field
- potential future directions
The conclusions were modified and structured according with your observation.
The CLC program is a very useful tool, but it should be improved in the future, considering that since its inception and so far many things have improved significantly, such as: satellite image quality, frequency over time. Their coverage of the target area, the know-how of the researchers involved, etc. We consider that a multiscale approach of the land cover change issues in Europe, would be more appropriate for the present necessity of the society, so we think that it is necessary to:
- maintain the current scale 1: 100 000, in the construction of the CLC layers, in order to be able to compare new data with the layers from the previous years;
- add an supplementary layer 1: 25 000, in order to achieve more refined results in the future, which will be correlated with the new imaging possibilities and with the additional information available in 2020. On this new layer an update of methodology and of the CLC nomenclature is recommended.
Our study, in Romania, at intermediate scales (pseudo-LAU1 scale) proves that the anomalies of land use can be detected using transition matrixes and a qualitative mapping approach. In the Romanian case, the interpretation of the transformations concerning the arable land (CLC21) put an emphasis on the role played by the major economic actors investing in agriculture. The case of vineyards (CLC221) is more complicated because the transformations might have a double lead force – climate change and new economic specializations in some particular regions. The wet areas and water bodies’ case study (CLC4 and CLC5) show that some areas of natural land cover are subject to changes, even if it concerns areas with a sound management and protection, such as the Danube Delta Nature Reserve. We consider that many of the CLC changes anomalies might be explained by targeted analyses and case studies focusing on specific areas or land use categories, our final recommendation. This kind of analysis can be extrapolated to other regions. The main limitative aspect of our research, excluding the micro-data consistency regarding the land use in time, is related to the construction of the alternative pseudo-LAU1 geometry. Other methods of LAU2 aggregation (hierarchical clustering with spatial constraint) will produce a different framework that needs to be explored. The method we used for the delineation of land use anomalies is based on transition matrixes, without encompassing the probabilities of change from one chronological milestone to another.
The added value of this method relies both on a methodological transfer from the field of economic geography (theory of convergence) and on the more friendly interpretations for policy makers.
The case studies proved that the combination between natural and socio-economic factors creates changes that sometimes are counterintuitive, when they are reported to the basic assumptions that scientist made regarding the dynamics of land cover.
Avoid descriptive sentences such as:
Page 24, lines 735-737: This section of the article is organized as a set of recommendations, illustrating the perspective of a heterogeneous group of researchers (remote sensing, GIS, human geography and classical physical geography) and their expert opinion.
Page 25, lines 769-771: The relevant key findings of our research will be exposed in the final part of this section. They include: specifications regarding the limitations of our methodological approach, a possible continuation of our study in further research and some basic conclusions.
all the descriptive sentences were removed

Reviewer 2 Report
The manuscript analyzes the land-use and land-cover changes (LULCC) in Romania based on the upscaled Corine Land Cover data from 2006 and 2018. The paper is much more devoted to the analysis of the LUCC than to the proposal of the methodological approach, however, there is enough information to be replicated elsewhere. Thus, the paper deserves publication. Some minor comments are detailed below:
- The abstract is too long. The Abstract has 439 words while most of the journals restrict it into 200-250 words.
- Please, identify all abbreviations whenever they appear the first time in the text.
- The introduction should not begin with the objective of the study. I suggest moving the first sentence to the end of this Section. The Introduction section should start providing the key information of Corine Land Cover platform (legend, applications, key citations, etc.). Since many CLC classes are reported and discussed in this paper, a section at the end of the manuscript showing the legend of CLC would be very helpful. Some methodological details in this section should be also moved to Materials and Methods.
- Figure 1 shows the data regarding the number of inhabitants in each pseudo-LAU1 polygon. I don´t think this information needs to be repeated in the following figures (the maps are becoming too polluted).
Author Response
The manuscript analyzes the land-use and land-cover changes (LULCC) in Romania based on the upscaled Corine Land Cover data from 2006 and 2018. The paper is much more devoted to the analysis of the LUCC than to the proposal of the methodological approach, however, there is enough information to be replicated elsewhere. Thus, the paper deserves publication. Some minor comments are detailed below:
- The abstract is too long. The Abstract has 439 words while most of the journals restrict it into 200-250 words.
We modified the abstract and now has no more than 250 words
Abstract: During the last 30 years, the Romanian economy faced different challenges due to structural readjustments, crisis overcoming and globalization. The share of the primary and secondary sectors in the Gross Domestic Product strongly decreased, while the services took-off. The main objective for this study is to observe how these economic readjustments can be assessed and measured using the Corine Land Cover datasets from 1990, 2000, 2006, 2012 and 2018 (with special observation on the range 2006 and 2018 after Romania was included in European Union). Despite some of the methodological limitations (like the minimum surface change), the Corine Land Cover turned out to be a powerful tool and it allowed us to detect an intense correlation between the socio-economic and the structural trends in land use, in specific spatial contexts. The artificial surfaces are constantly increasing and this trend is rather visible as a distance function to the major Romanian cities. The most interesting changes occurred in the case of the agricultural polygons. The main trend emphasized by our analysis regards the redeployment of large farms in areas of agronomic and environmental territorial optimum. Such is the case for vineyards (after a decline during the 2000-2006) and for annual cultures. All these changes in the land use patterns are too complex to be encompassed by a single methodology, that’s why we used different tools, ranging from spatial analysis to geo-economic modeling, in order to detect how the Corine Land Cover datasets might be used for a better understanding of the Romanian economic readjustments.
- Please, identify all abbreviations whenever they appear the first time in the text.
We explain all abbreviations where they appear the first time in the text.
- The introduction should not begin with the objective of the study. I suggest moving the first sentence to the end of this Section. The Introduction section should start providing the key information of Corine Land Cover platform (legend, applications, key citations, etc.). Since many CLC classes are reported and discussed in this paper, a section at the end of the manuscript showing the legend of CLC would be very helpful. Some methodological details in this section should be also moved to Materials and Methods.
All the introduction part was modified as follows:
After 1990, during the post-communist period, Romania passed through complex social, economic and political transformations which lead to radical background changes within some important fields such as land property and agricultural land exploitation. Some of these transformations were investigated at global [1], European [2] and national level using Corine Land Cover (CLC) datasets [3, 4, 5]. The CLC project largely fulfills the expectations of the researchers and other users [6], having two main advantages: it provides a seamless geometry usable for macro-regional studies and an internal classification of the land use/cover categories that allows time traceability of changes, for a reasonable period (1990-2018). The facility of tracking the changes in the use of the land has allowed a multitude of applications and correlations between CLC datasets and different parameters [7, 8, 9]. There is an intense use of the CLC datasets for local or regional studies; however, the intermediate scales are systematically neglected by other researches [10, 11, 12, 13]. That’s why we consider that the proper scale of CLC use for policy design should be an intermediate one, an administrative geometry that fits the needs of analysis somewhere between the local level and the NUTS3 delineation [14].
The use of CLC at local administrative unit (LAU2) is largely unadvised [15], as the degree of land use generalization might interfere with the local patterns. In this case, numerous policy makers will eventually use CLC just as an informative base of interpretation and not as an analytical frame. At NUTS3 scale, the CLC dataset overlaps national or regional statistical information that, in some conditions, creates data redundancy or impossible comparisons (mainly in the case of the artificial surfaces) [16].
In this paper we try to investigate the land cover and land changes in Romania, using an appropriate scale of analysis – an intermediate one between the main administrative frame and the local scale. The period we emphasized in our research covers mainly the 2006-2018 time intervals. The methodology we developed is applied at a national scale, but it can be replicated for other states too.
The problem of the intermediate scale of analysis is mainly related to its construction. If the studies at local scale generally involve a limited set of beneficiaries (policy makers), an intermediate scale would mean to aggregate administrative polygons belonging to a larger set of decision takers. One of the intentions of our study was to develop a method of administrative polygons aggregation that can be intersected with the CLC datasets. This intermediate scale of analysis was labeled pseudo-LAU1, where LAU1 stands for local administrative units of first rank. Using spatial analysis techniques and potential accessibility functions [17], was created an operational pseudo-LAU1 geometry that was used in order to collect the land-use and land-cover information provided in the CLC vector database (2006-2018). The building blocks of this operational geometry are the Romanian local administrative units known as LAU2.
The detection of the anomalies in the land-use/cover changes that we propose in this research has its background in a methodological import from the theory of the economic convergence, more specifically the elaboration of a dynamic typology using class transition matrixes [18]. The integration of the CLC indicators in the alternative geometry allows the evaluation of the relative share of each land-use category in a double chronological context (time t0= 2006 and time t1=2018). If a homogeneous classification in time is applied on the data, the possible state of a pseudo-LAU1 spatial unit takes three possible values: stagnation, positive evolution or decline. Strong evolutions between classes can be assimilated to spatial anomalies in the dynamics of different CLC indicators and they can provide a basis for the selection of interesting case studies [19]. The validation and explanation of the eventual anomalies detected by the class transition matrix is realized using specific geo-statistical techniques and remote sensing analysis. The case studies are declined on artificial surfaces, on a selected set of agricultural land-use dynamics (arable land, vineyards), on forests and on some of the natural land cover layers (wetlands and water bodies). For a better contextual understanding of the main trends registered in the land-use dynamics in Romania, supplementary statistical data was collected (new artificial surfaces in rural areas – mainly agro-industrial facilities) and a chronological dataset of the CLC changes that was created as a data quality check tool. From a policy perspective, a sound assessment of land use changes and anomalies in different territorial context is a topic of major interest, as the land use dynamics might interfere with targeted policies applied for the rural development, urban sprawl, transportation and environment.
Considering the ones described above the principal objectives of this article is to investigates potential ways in which the CLC dataset can be used for more in depth analysis of land-use and land cover transformations, at an intermediate scale, appropriate for policy design and decision process in Romania.
At the end of the paper we introduce a legend of Corine Land Cover
Class 1. Artificial surfaces include: 1.1. Urban fabric (1.1.1. Continuous urban fabric, 1.1.2. Discontinuous urban fabric); 1.2. Industrial, commercial and transport units (1.2.1. Industrial and commercial units, 1.2.2. Road and rail networks and associated land, 1.2.3. Port areas, 1.2.4. Airports); 1.3. Mine, dump and construction sites (1.3.1. Mineral extraction sites, 1.3.2. Dump sites, 1.3.3. Construction sites); 1.4. Artificial non-agricultural vegetated areas (1.4.1. Green urban areas, 1.4.2. Sport and leisure facilities)
Class 2. Agricultural areas include: 2.1. Arable land (2.1.1. Non-irrigated arable land, 2.1.2. Permanently irrigated land, 2.1.3. Rice fields); 2.2. Permanent crops (2.2.1. Vineyards, 2.2.2. Fruit trees and berry plantations, 2.2.3. Olive groves); 2.3. Pastures (2.3.1. Pastures), 2.4. Heterogeneous agricultural areas (2.4.1. Annual crops associated with permanent crops, 2.4.2. Complex cultivation patterns, 2.4.3. Land principally occupied by agriculture, with significant areas of natural vegetation, 2.4.4. Agro-forestry areas)
Class 3. Forests and semi-natural areas include: 3.1. Forests (3.1.1. Broad-leaved forest, 3.1.2. Coniferous forest. 3.1.3. Mixed forest); 3.2. Shrub and/or herbaceous vegetation association (3.2.1. Natural grassland, 3.2.2. Moors and heathland, 3.2.3. Sclerophyllous vegetation, 3.2.4. Transitional woodland shrub); 3.3. Open spaces with little or no vegetation (3.3.1. Beaches, dunes, and sand plains, 3.3.2. Bare rock, 3.3.3. Sparsely vegetated areas, 3.3.4. Burnt areas, 3.3.5. Glaciers and perpetual snow)
Class 4. Wetlands include: 4.1. Inland wetlands (4.1.1. Inland marshes, 4.1.2. Peatbogs); 4.2. Coastal wetlands (4.2.1. Salt marshes, 4.2.2. Salines, 4.2.3. Intertidal flats)
Class 5. Water bodies include: 5.1. Inland waters (5.1.1. Water courses, 5.1.2. Water bodies); 5.2. Marine waters (5.2.1. Coastal lagoons, 5.2.2. Estuaries, 5.2.3. Sea and ocean)
- Figure 1 shows the data regarding the number of inhabitants in each pseudo-LAU1 polygon. I don´t think this information needs to be repeated in the following figures (the maps are becoming too polluted).
We respect this observation and made the changes in the figures to be more clear
Thank you for your clearly observation. They helped us a lot to correct and to restructure the whole work.

Reviewer 3 Report
As a sample of a reader, I felt difficult to read and understand this paper. It may be partly because of my lack of knowledge, but I wished the following things:
- Simpler language, please. Each sentence is a little complicated. It should be stated simpler, shorter, and more directly.
- Shorter, please. 27 pages of paper is too long. The abstract is also too long. It can be more concise, perhaps.
- More introductory explanation, please. I am interested in remote sensing of landcover. But this paper is very much about land use study after remote sensing. Therefore, more explanations about ideas and words about land use study may be necessary.
- More consistency, please. Use of symbols, units, and formula are not self-consistent of consistent with conventional rules.
Because these difficulties were so much for me, I am afraid but I gave up reading this paper. Overall, I felt this paper is not suitable in this journal. I wish the authors to publish this paper in a more specialized journal in land use study or rewrite it so as to fit with the readers of this journal.
Specific comments:
1. Line 17 "with various spatial print" ... I do not understand.
2. Line 21 "leaded" ... perhaps "led".
3. Line 35-39 ... I do not understand.
4. Line 63 ... perhaps a grammar mistake.
5. Line 86 "time t=2006 and time t+1=2018" ... strange. It means "1=2". Maybe "time t0=2006 and t1=2018" or something like that.
6. Line 106 "quit" ... perhaps "quite"
7. Line 113 " 25HA/100M" ... Please use conventional unit symbols of SI. HA should be ha. M should be m. These are specified in an SI document (P131 and P145):
https://www.bipm.org/en/publications/si-brochure/
8. Line 156, the equation. ... "i" of Pi must be subscript. Same for Mi, Mj, and Dij. "-1" on Dij may not be necessary (please check).
9. Line 157 "LAU2 I;" ... Perhaps not I but i.
10. Line 15-159 "Pi", "Mi", "Dij" ... The symbols should be not in roman but in italic. Please check the SI document above.
11. Line 160 "separatesi" ... "separates i".
12. Line 207-208 "t=2006 and t+1=2018" ... same situation with 5. Also in Table 1.
13. Table 2. "r", "p" ... The symbols should be not in roman but in italic.
14. Line 417 and 420. "mil." ... perhaps million or mega. "mil." is not advised in the SI document. It is confusing with "mili".
15. Line 508-510. "The modification of the surface ... aquatic units." ... Can you rewrite it simpler and shorter? The meaning of the sentence is so simple. This is just one example. By using simpler, shorter, and more direct words, you can reduce the frustrations of readers.
16. Figures 14 and 15 ... It may be better to put national borders on the maps.
Author Response
As a sample of a reader, I felt difficult to read and understand this paper. It may be partly because of my lack of knowledge, but I wished the following things:
- Simpler language, please. Each sentence is a little complicated. It should be stated simpler, shorter, and more directly.
We tried to simplify the text by using a proofreading expertise for English language.
- Shorter, please. 27 pages of paper is too long. The abstract is also too long. It can be more concise, perhaps.
After modification we made in the text and figures all paper was reduced to 24 pages
- More introductory explanation, please. I am interested in remote sensing of landcover. But this paper is very much about land use study after remote sensing. Therefore, more explanations about ideas and words about land use study may be necessary.
We made modification in the introduction part to be more concise and connected with international researches made in the field of land cover.
- More consistency, please. Use of symbols, units, and formula are not self-consistent of consistent with conventional rules.
we made all the modifications required for symbols, units and formula.
Because these difficulties were so much for me, I am afraid but I gave up reading this paper. Overall, I felt this paper is not suitable in this journal. I wish the authors to publish this paper in a more specialized journal in land use study or rewrite it so as to fit with the readers of this journal.
Specific comments:
- Line 17 "with various spatial print" ... I do not understand.
Because the abstract was too long we reduce a part of the initial text to be more concise.
Line 21 "leaded" ... perhaps "led".
Because the abstract was too long we reduce a part of the initial text to be more concise.
Line 35-39 ... I do not understand.
Because the abstract was too long we reduce a part of the initial text to be more concise.
Line 63 ... perhaps a grammar mistake.
all the text from this part of introduction was changed (see the new text in red).
Line 86 "time t=2006 and time t+1=2018" ... strange. It means "1=2". Maybe "time t0=2006 and t1=2018" or something like that.
thank you for this observation t0=2006
Line 106 "quit" ... perhaps "quite"
we made the change
Line 113 " 25HA/100M" ... Please use conventional unit symbols of SI. HA should be ha. M should be m. These are specified in an SI document (P131 and P145):
https://www.bipm.org/en/publications/si-brochure/
We made in all text the changes about conventional unit symbol
Line 156, the equation. ... "i" of Pi must be subscript. Same for Mi, Mj, and Dij. "-1" on Dij may not be necessary (please check).
We made the changes in the equation form.
Line 157 "LAU2 I;" ... Perhaps not I but i.
we made the changes in the text.
Line 15-159 "Pi", "Mi", "Dij" ... The symbols should be not in roman but in italic. Please check the SI document above.
we made all the changes
Line 160 "separatesi" ... "separates i".
we also made the changes required
Line 207-208 "t=2006 and t+1=2018" ... same situation with 5. Also in Table 1.
we also made the changes required
Table 2. "r", "p" ... The symbols should be not in roman but in italic.
we put the simbols in italic
Line 417 and 420. "mil." ... perhaps million or mega. "mil." is not advised in the SI document. It is confusing with "mili".
we change in all text mil with million
Line 508-510. "The modification of the surface ... aquatic units." ... Can you rewrite it simpler and shorter? The meaning of the sentence is so simple. This is just one example. By using simpler, shorter, and more direct words, you can reduce the frustrations of readers.
we modified the phrase in: The modifications of the surfaces included in the category of wetlands and those covered by water are induced by a series of anthropic and natural mechanisms.
Figures 14 and 15 ... It may be better to put national borders on the maps.
because this maps was not very clear we decide to delete them from the paper
Thank you for your clearly observation. They helped us a lot to correct and to restructure the whole work.

Round 2
Reviewer 1 Report
I am satisfied with the responses of the authors and their revision. Extensive changes were made in the structure of manuscript. The content of chapters was significantly shortened and clarified.
Author Response
Dear Reviewer thank for your comments and observation. we tried to fulfill al your requirements.